# Flexible neural population dynamics govern the speed and stability of sensory encoding in mouse visual cortex

Edward A. B. Horrocks ◉[1] ✉, Fabio R. Rodrigues ◉[1] & Aman B. Saleem ◉[1] ✉

Time courses of neural responses underlie real-time sensory processing and perception. How these temporal dynamics change may be fundamental to how sensory systems adapt to different perceptual demands. By simultaneously recording from hundreds of neurons in mouse primary visual cortex, we examined neural population responses to visual stimuli at sub-second time-scales, during different behavioural states. We discovered that during active behavioural states characterised by locomotion, single-neurons shift from transient to sustained response modes, facilitating rapid emergence of visual stimulus tuning. Differences in single-neuron response dynamics were associated with changes in temporal dynamics of neural correlations, including faster stabilisation of stimulus-evoked changes in the structure of correlations during locomotion. Using Factor Analysis, we examined temporal dynamics of latent population responses and discovered that trajectories of population activity make more direct transitions between baseline and stimulus-encoding neural states during locomotion. This could be partly explained by dampening of oscillatory dynamics present during stationary behavioural states. Functionally, changes in temporal response dynamics collectively enabled faster, more stable and more efficient encoding of new visual information during locomotion. These findings reveal a principle of how sensory systems adapt to perceptual demands, where flexible neural population dynamics govern the speed and stability of sensory encoding.

Neural responses to sensory inputs fluctuate at sub-second time-scales and these temporal dynamics underlie sensory processing and perception[1–4]. How temporal response dynamics change may therefore be fundamental to how sensory systems adapt to different perceptual demands. For example, when moving through an environment, visual inputs can change rapidly[5], necessitating faster neural processing for more immediate behavioural responses[6,7]. Indeed, during locomotion mice can discriminate the direction of moving stimuli within 300ms[8], highlighting the importance of sub-second timescale temporal response dynamics in visual perception. Whilst previous research has identified various mechanisms within the mouse visual system that generally enhance the encoding of sensory

inputs during locomotion[6,7,9–18], heightened arousal[11,19–23] and spatial attention[18,24,25], these studies have tended to focus on second(s)-long trial spike counts, leaving fundamental gaps in our understanding of real-time sensory processing[7,9,10,16,22,26,27].

Sensory neurons can respond to external stimuli with varied temporal dynamics, often exhibiting transient onset responses which later settle into a sustained response[2,3,28,29]. These temporal dynamics can be crucial for sensory processing[4], but how temporal response dynamics and resultant stimulus encoding can change during different behavioural states remains understudied. One hypothesis is that differences in responses and stimulus encoding between behavioural states are constant over time, a time-invariant response modulation.

[1]Institute of Behavioural Neuroscience, University College London, London WC1V 0AP, UK. ✉e-mail: edward.horrocks.17@ucl.ac.uk; aman.saleem@ucl.ac.uk

Alternatively, differences in responses between states may vary over time and exhibit altered temporal dynamics. Here, we hypothesised that during locomotion the temporal dynamics of responses should change to facilitate faster neural processing of behaviourally-relevant sensory inputs.

Sensory information is also encoded through the coordinated activity of populations of neurons, which can be characterised using pairwise correlations[30]. As neural correlations can vary at fast timescales[31,32], changes in their temporal dynamics may be critical to the real-time encoding of sensory inputs[31]. Notably, noise correlations of spike counts in second(s)-long trials are reduced during locomotion and states of heightened arousal in mouse primary visual cortex (V1)[7,10,11,18], and this decorrelation of population responses is thought to improve the encoding of visual stimuli[7,11]. Yet how stimulus-evoked neural correlations can vary across time and influence sensory encoding during different behavioural states is not well established. Given that neural correlations can be indicative of functional connectivity[30,32–35], changes in their dynamics could also provide insight into how stimulus-evoked functional connectivity evolves over time to shape population responses during different behavioural states[27,36–39]. As both stimulus encoding by correlated population activity and functional connectivity between neurons are believed to be strongly dependent on behavioural state[7,11,27,39,40], we hypothesised that the temporal dynamics of pairwise correlations should be distinct in stationary and locomotion states to reflect this. Functionally, we predicted that changes in pairwise correlation dynamics during locomotion should support rapid neural population-level encoding of visual stimuli.

Latent variable models of large-scale neural recordings enable investigation into population-level computations[41–48]. In particular, these methods have afforded a dynamical systems approach where population activity is treated as a time-evolving multidimensional variable[41,44,49,50], and provide insight into the intrinsic neural population dynamics that shape sensory responses[49]. Understanding how intrinsic neural population dynamics can change during different behavioural states, and in turn how these dynamics shape population responses to sensory inputs, should provide key insight into how sensory systems implement flexible sub-second timescale sensory processing.

Here, using large-scale electrophysiology (4-shank Neuropixel 2.0 probes[51]), we analysed the neural responses of 100 s of simultaneously recorded neurons in mouse primary visual cortex (V1) at fast, behaviourally-relevant timescales (10 s of ms). Our findings reveal changes in the temporal dynamics of single-neuron firing rate responses, correlations between neurons and latent trajectories of population activity between stationary and locomotion behavioural states. Functionally, changes in temporal response dynamics during locomotion enable faster, more stable and more efficient integration of new visual information into ongoing neural population activity. More generally, our findings establish a principle of how neural population dynamics flexibly adapt during different behavioural states to govern the speed and stability of sensory encoding.

## Results

We investigated the temporal dynamics of neural responses recorded from mouse V1 during different behavioural states using large-scale electrophysiology recordings (4-shank Neuropixel 2.0 probes[51]; Fig. 1a, b; $n = 5$ mice; $n = 1583$ 'good' units). Individual recording sessions consisted of hundreds of neurons (mean ± SEM = 317 ± 40 'good' units per session) spanning ~820 um mediolaterally and ~700 um dorsoventrally in mouse V1 (with the exception of one recording which was performed with a single-shank probe). Mice were head-fixed and free to run on a polystyrene wheel while we presented dot field stimuli that moved in the naso-temporal direction with one of six visual speeds (0, 16, 32, 64, 128, 256°/s; Fig. 1a). Stimuli were presented on a truncated dome[52] and covered a large portion of the visual field (−120°

to 0° azimuth and −30° to 80° elevation). Each stimulus lasted for 1 s, with a 1-s grey screen inter-stimulus interval. We analysed neural responses to visual stimuli while mice were in stationary and locomoting states. Trials, which we defined as 200 ms pre-stimulus onset to 800 ms post-stimulus offset, were classified as locomotion if mean locomotion speed was >3 cm/s and remained >0.5 cm/s for >75% of the trial. Trials were classified as stationary if mean locomotion speed was <0.5 cm/s and remained <3 cm/s for >75% of the trial.

### Single-neuron temporal dynamics are less transient during locomotion

Single-neuron responses had more reliable temporal dynamics during locomotion. We characterised the temporal dynamics of single-neuron responses based on the shapes of their smoothed peri-stimulus time histograms (PSTHs) of spiking activity. We first found that during locomotion, responses were more likely to have reliable shapes across repeated trials of the same stimuli (Stationary: 22% of responses reliable; Locomotion 30%; $p < 0.001$ McNemar test for difference in paired proportions; $n = 9498$ paired responses from 1563 units and 6 stimuli; 19% of responses were reliable in both stationary and locomotion trials). This increased reliability is consistent with less variable subthreshold visually-evoked responses observed during locomotion[12,53]. Having established that responses tend to be more reliable during locomotion, we asked whether behavioural state simply scales the firing rates of PSTHs or if it also alters their shape? To answer this, we compared the temporal dynamics of reliable responses recorded in stationary and locomotion trials.

Single-neuron responses had less transient stimulus onset dynamics during locomotion. During stationary trials, single neurons typically responded to visual stimulus onset with a large transient increase in firing rate which quickly decreased, before eventually settling to a lower steady-state rate (Fig. 1c–e and Supplementary Fig. 1). By contrast, during locomotion trials firing rates tended to either increase directly to the steady-state rate or exhibit a more gradual decline from their initial peak. This was true even when responses had similar steady-state firing rates in stationary and locomotion trials (Fig. 1c−2nd example). We also found comparable differences in visual response dynamics following stimulus offset. In stationary trials, responses often exhibited a transient decrease in firing rate below the baseline rate following stimulus offset, whilst in locomotion trials this decrease was either reduced or absent (Fig. 1c–e and Supplementary Fig. 1). We quantified these observations by fitting descriptive functions (*Decay*, *Rise*, *Peak*, *Trough* or *Flat*; Fig. 1f) separately to the stimulus onset ($t = 0–0.3$ s) and offset ($t = 1–1.5$ s) periods of each reliable response, using the best-fitting function to classify their stimulus onset and offset features. Responses from locomotion trials were significantly less likely to have *Peak*-onset features and instead were more likely to have *Rise*-onset features (Fig. 1g; both $p < 0.001$, Generalised Linear Mixed-Effects (GLME) model, effect of behavioural state; stationary: $n = 2077$ reliable responses; locomotion: $n = 2802$), reflecting their less transient stimulus onset dynamics. During stimulus offset, responses from locomotion trials were significantly less likely to have *Trough*- or *Flat*-offset features (both $p < 0.001$) but more likely to have *Decay*-offset features (Fig. 1g; $p < 0.001$), reflecting a steadier decline in firing rate following stimulus offset.

Single-neuron responses were more sustained during locomotion. Given the less transient features we observed in responses recorded in locomotion trials, we quantified how sustained responses were overall by calculating a sustainedness index that measures the ratio of the baseline-corrected mean and peak firing rates. The index converges to 1 as the mean firing rate approaches the peak firing rate and converges to 0 for large differences between mean and peak firing rates (Fig. 1h). Responses were significantly more sustained in locomotion trials (Fig. 1h and Supplementary Fig. 1; Median (IQR) in stationary trials: 0.32 (0.18–0.48); locomotion trials: 0.48 (0.32–0.62); $p < 0.001$ Linear

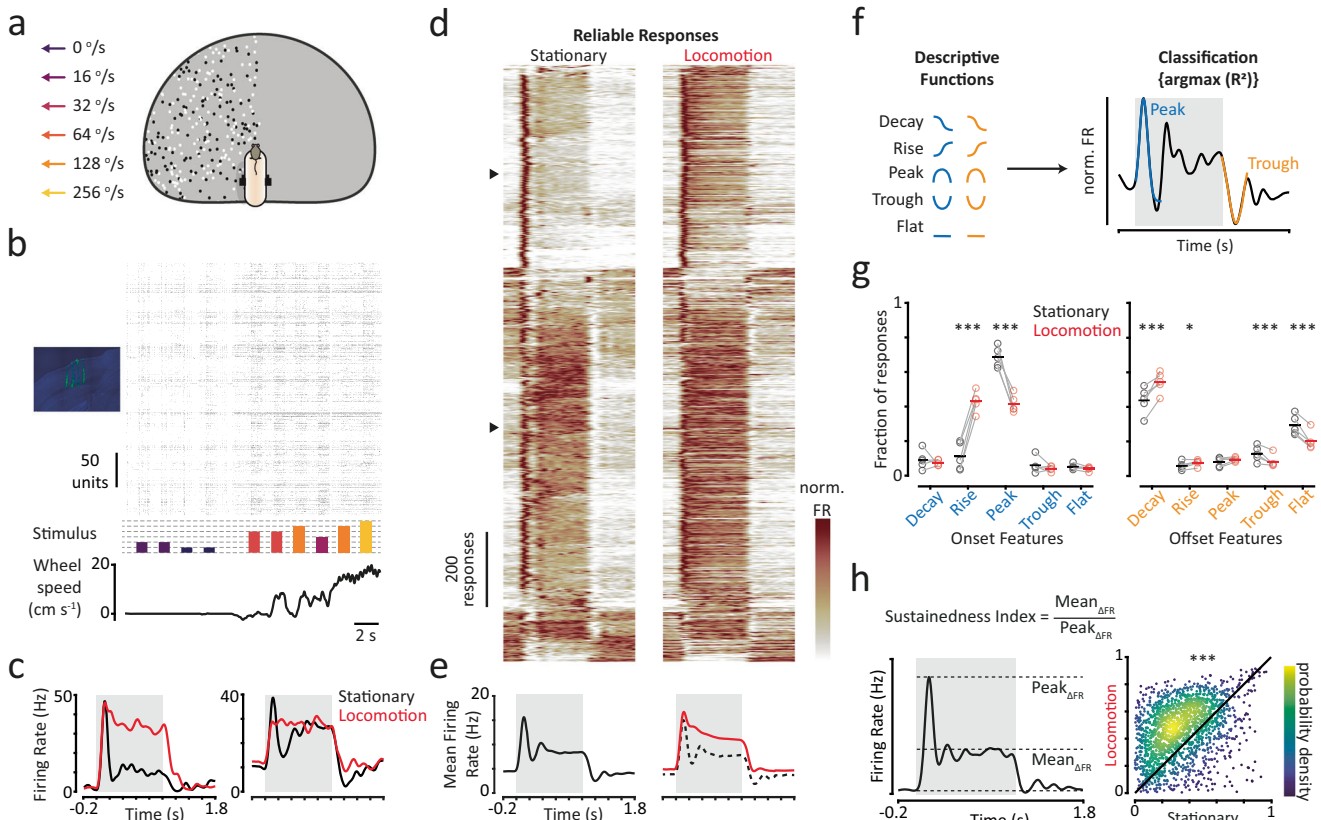

**Fig. 1 | Single-neuron responses are less transient during locomotion. a** Mice were head fixed on a wheel while dot field stimuli moving at one of six visual speeds were presented. Stimuli were presented on the interior surface of a truncated dome and covered a large portion of the contralateral visual field. **b** We recorded from monocular V1 using neuropixel 2.0 probes and analysed responses while mice were in stationary and locomotion states. **c** Smoothed, trial-averaged responses (PSTHs) from stationary and locomotion trials, to the same stimulus speed, for two example neurons. Grey shaded region indicates the stimulus period. **d** All responses that were reliable in both stationary and locomotion trials. Each row is a pair of responses from stationary and locomotion trials to the same stimulus speed from the same neuron. Responses are sorted based on stationary responses using a hierarchical method. Firing rates are normalised (min–max scaling). Triangles correspond to examples in (**c**). **e** Mean reliable responses from stationary (left) and locomotion (right) trials. Dashed line in the right panel indicates the mean reliable

stationary response for comparison. **f** Overview of response characterisation using descriptive functions to define stimulus-onset and -offset features. **g** Proportions of reliable responses classified with different onset and offset features. Individual data points are paired proportions of responses from each session for stationary (black) and locomotion (red) trials. Horizontal lines represent the fraction of all reliable responses with a given feature. Generalised Linear Mixed-Effects (GLME) model analysis, two-sided effect of behavioural state ($n = 2077$ reliable responses during stationary states, $n = 2802$ reliable responses during locomotion states). Adjustments were not made for multiple comparisons. *$p < 0.05$; **$p < 0.01$; ***$p < 0.001$ (see Supplementary Information for exact $p$ values). **h** Left panel: overview of sustainedness index. Right panel: scatter density plot of sustainedness index values for paired reliable responses recorded during stationary and locomotion trials. GLME analysis, two-sided effect of behavioural state. ***$p = 5.33 \times 10^{-103}$. Colorbar scale is arbitrary. Source data are provided as a Source Data file.

Mixed-Effects (LME) model, effect of behavioural state; $n = 1807$ paired responses that were reliable in stationary and locomotion trials). This was because while both baseline-corrected mean and peak firing rates increased in locomotion trials, mean firing rates increased significantly more (Median (IQR) fractional change in mean firing rate: 2.03 (1.10–4.07); peak firing rate: 1.17 (0.73–1.72); $p < 0.001$ Sign-rank test; $n = 1807$ paired responses that were reliable in stationary and locomotion trials).

The changes in response dynamics we observed were also comparable across cell types and visual brain areas. We found that these state-dependent changes in response dynamics are similar across different cell-types defined based on electrophysiological characteristics[54] (Supplementary Fig. 2). We also found comparable changes in single-neuron temporal dynamics between stationary and locomotion states across visual cortical (V1, LM, AL, RL, PM, AM) and thalamic (LGN, LP) areas, based on our analysis of the Allen Institute's 'Visual Coding' dataset[55] (Supplementary Fig. 3).

Collectively, these results show that the temporal dynamics of single-neuron responses can vary between behavioural states. During locomotion, single-neuron responses are more reliable, have less

transient stimulus onset and offset dynamics, and have more sustained changes in firing rate. The effects of behavioural state on single-neuron responses are therefore time-varying and cannot be explained solely by a scaling of firing rates.

Changes in the temporal dynamics we observed with behavioural state could not be explained by other behavioural factors. Differences in response dynamics were robust to different criteria for defining behavioural states (Supplementary Fig. 4) or removing trials with eye movements (Supplementary Fig. 5). Response dynamics did not vary substantially between slow and fast locomotion speeds (Supplementary Fig. 6). However, stationary trials with higher levels of pupil-indexed arousal did produce weak effects on response dynamics compared to locomotion trials (Supplementary Fig. 5), potentially reflecting the intermediate nature of this state between high arousal (locomotion state) and low arousal (stationary and constricted pupil).

**Tuning for visual speed begins earlier and persists for longer during locomotion**

After establishing how single-neuron temporal dynamics can adapt to different behavioural states, we next investigated how these changes

affect visual stimulus encoding. We analysed how 'tuned' responses were, that is, how well they distinguished stimuli moving at different visual speeds. We hypothesised that tuning for visual speed should emerge earlier during locomotion to meet the more immediate perceptual demands during this state. We quantified the strength of visual speed tuning across different time windows (200 ms sliding window; 10 ms step size) using the cross-validated coefficient of determination ($R^2$), which determines how reliable a tuning curve is across trial repeats. This tuning strength metric takes values between −1 and 1, with values greater than 0 indicating some degree of tuning for visual speed. We classified bouts of visual speed tuning as periods where tuning strength was both significant (using a shuffle control) and exceeded a threshold of $R^2 \geq 0.1$ for at least five consecutive times points (≥50 ms step size). We considered a neuron to be tuned if it had at least one bout of visual speed tuning. Using this metric, we found that more neurons were tuned during locomotion trials (Stationary 26%; Locomotion: 38%; $p < 0.001$ McNemar test; $n = 1583$ units). Next, to investigate how tuning varied over the time course of responses between behavioural states we considered only neurons that were tuned in both stationary and locomotion trials ($n = 344$ units, 22%).

Tuning for visual speed started earlier and persisted for longer during locomotion. Whilst tuning for visual speed tended to emerge slowly and strengthen gradually over time in stationary trials, it often emerged rapidly following stimulus onset in locomotion trials (Fig. 2a–c). Overall, tuning started almost twice as early in locomotion

trials, a difference that was largely caused by an increase in the number of neurons becoming tuned during the initial 200 ms following stimulus onset (Fig. 2d; median tuning start times in stationary trials: 180 ms; locomotion trials: 100 ms. $p < 0.001$ LME model, effect of behavioural state; $n = 344$ units). Tuning also finished significantly later in locomotion trials, reflecting a tendency for neurons to remain tuned for more prolonged periods following stimulus offset (Fig. 2e; median tuning finish times in stationary trials: 1070 ms; locomotion trials: 1150 ms; $p < 0.001$). Strikingly, neurons were on average tuned for twice as long in locomotion trials (Fig. 2f; median tuning durations in stationary trials: 465 ms; locomotion trials: 955 ms; $p < 0.001$). This improved tuning for visual speed during locomotion was associated with a large increase in the dynamic range of mean responses to different visual speeds (Fig. 2g; Mean ± SEM over the stimulus period: stationary = 2.17 ± 0.11 Hz; locomotion = 4.41 ± 0.19 Hz; $p < 0.001$; LME model, effect of behavioural state; $n = 1583$ units).

Improvements in visual speed decoding during locomotion were greatest immediately following stimulus onset. Improved visual speed tuning enhanced our ability to decode visual speed using a linear decoder that assumed independent neurons (Fig. 2h; $p < 0.001$ mixed-model ANOVA effect of behavioural state; $p < 0.001$ significant interaction between behavioural state and time; $n = 5$ subjects). Decoding performance in stationary trials initially increased following stimulus onset before dipping at ~200 ms, corresponding to the transient decrease in firing rates we observed during this period

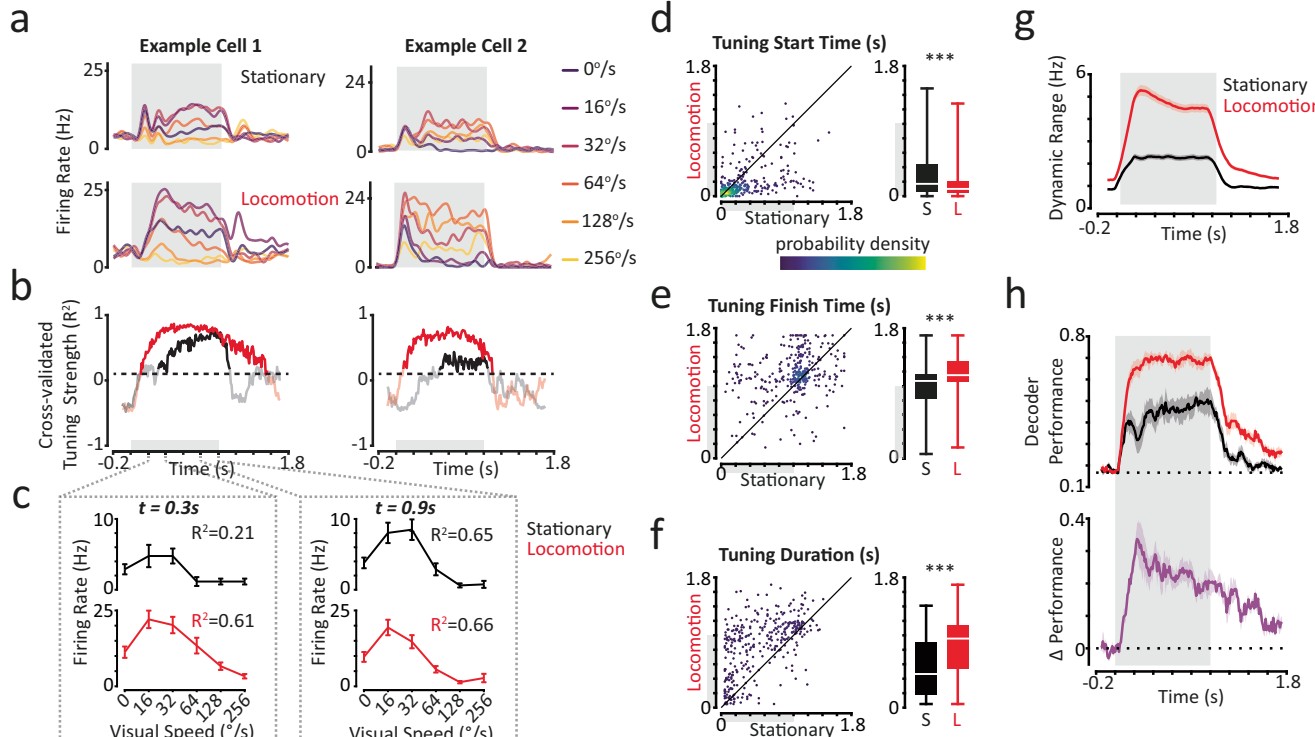

**Fig. 2 | Tuning for visual speed begins earlier and persists for longer during locomotion. a** Sets of responses to each visual speed presented during stationary and locomotion trials for two example neurons. Grey shaded region indicates stimulus period. **b** Tuning strength profiles over the response period (200 ms sliding window; 10 ms step size) of the example cells in (**a**), for stationary (black) and locomotion (red) trials. The dashed line indicates the threshold for determining when neurons were tuned. **c** Visual speed tuning curves obtained from example cell 1 at two different time points, corresponding to the dashed lines in (**b**). Data are presented as mean ± SEM across trial repeats ($n = 33$ per condition). Tuning strength ($R^2$) are noted for each time point. **d** Tuning start times for all neurons with a valid period of tuning in both stationary and locomotion trials ($n = 344$ cells). Left panel: scatter density plot of tuning start times in stationary and locomotion trials.

Colorbar scale is arbitrary. Right panel: Box plots of tuning start times. (S Stationary, L Locomotion). Centre white lines indicate medians, box limits indicate upper and lower quartiles and whiskers indicate full range of data. Linear Mixed Effects (LME) model analysis, two-sided effect of behavioural state. ***$p = 3.36 \times 10^{-11}$. **e** Same as (**d**) for tuning finish times. ***$p = 1.65 \times 10^{-13}$. **f** Same as (**d**) for tuning durations. ***$p = 6.42 \times 10^{-46}$. **g** Mean dynamic range of single-neuron sets responses to the six visual speeds presented during stationary and locomotion trials. **h** Top panel: session-mean decoding performance (regularised LDA with a diagonal covariance matrix) over time (100 ms sliding window; 10 ms step size) for stationary and locomotion trials. Bottom panel: corresponding session-mean difference in decoding performance between locomotion and stationary trials. Shaded regions indicate mean ± SEM across subjects ($n = 5$). Source data are provided as a Source Data file.

(Figs. 1c–g and 2a). Subsequently, performance gradually increased over the stimulus period. By contrast, decoding performance in locomotion trials increased rapidly to a higher sustained level following stimulus onset. As a result, improvements in visual speed decoding during locomotion were not uniform over time but were greatest ~200 ms following stimulus onset, gradually declining over the remainder of the response period (Fig. 2h, bottom panel). The speed of visual stimulus encoding by single-neurons in mouse V1 is therefore dependent on behavioural state.

### Changes in the dynamics of pairwise neural correlations support more efficient stimulus encoding during locomotion
Having established how single-neuron temporal dynamics and stimulus tuning can be affected by behavioural state, we next focused on the temporal dynamics of co-ordinated neural population responses. We focused initially on the dynamics of pairwise neural correlations which characterise functional interactions within a population of neurons. From a stimulus encoding perspective, neural correlations can be partitioned into *signal* correlations, which are associated with correlated average responses to a varying feature of

interest (in this case visual speed), and *noise* correlations, which reflect all other trial-by-trial sources of correlation[30]. We observed a range of signal and noise correlations between individual pairs of neurons whose dynamics could vary substantially between stationary and locomotion trials (Fig. 3a). We assessed how average signal and noise correlations varied across the response period during stationary and locomotion states.

Signal correlations strengthened and noise correlations weakened during locomotion. The magnitude of signal correlations increased almost twofold during locomotion trials, while maintaining a similar time course to stationary trials (Fig. 3b, Mean ± SEM absolute value over the stimulus period: stationary = 0.044 ± 0.000; locomotion = 0.0767 ± 0.001; $p < 0.001$; LME model, effect of behavioural state). This is consistent with the increased separation of mean single-neuron responses to different visual speeds during locomotion, which was evident in our estimates of dynamic range (Fig. 2g). Conversely, the magnitude of noise correlations was significantly reduced overall during locomotion (Fig. 3b; stationary = 0.096 ± 0.001; locomotion = 0.0607 ± 0.001; $p < 0.001$), in agreement with previous reports of reduced second(s)-long spike-count noise correlations[7,10,11]. This was

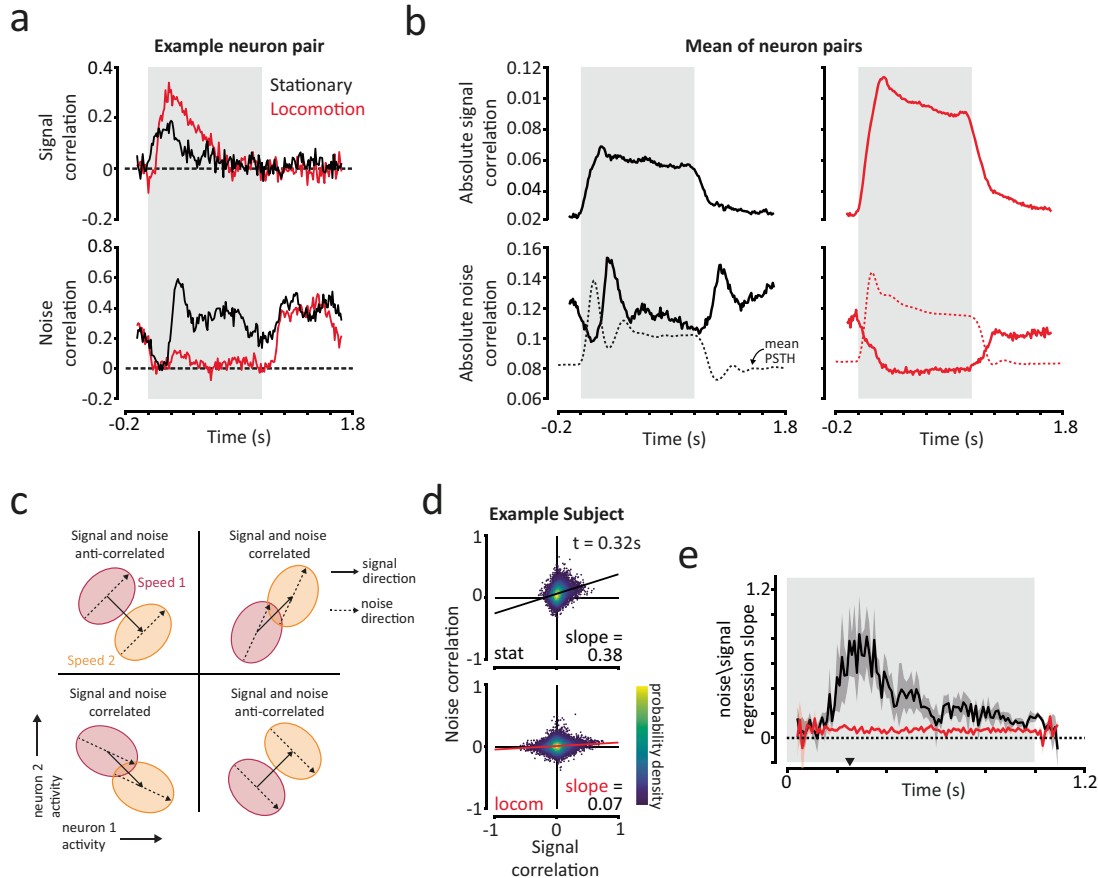

**Fig. 3 | Changes in pairwise neural correlation dynamics increase stimulus encoding capacity during locomotion. a** Signal and noise correlations over the response period (200 ms sliding window; 10 ms step size) for an example neuron pair during stationary and locomotion trials. Grey shaded region indicates stimulus period. **b** Mean absolute signal (top panels) and noise (bottom panels) correlations for stationary and locomotion trials. Mean reliable PSTHs are plotted as dashed lines in bottom panels for reference. **c** Overview of how different relationships between signal and noise correlations influence stimulus encoding for a neuron pair (principles hold for larger numbers of neurons). Each ellipse represents the distribution of responses for a specific stimulus. Stimulus encoding can be assessed by the degree of overlap between the ellipses. Noise correlations are detrimental to

stimulus encoding when correlated with signal correlations but can improve stimulus encoding when anti-correlated with signal correlations. **d** Scatter density plots of the relationship between signal and noise correlations for an example subject at a single time point, for stationary (top) and locomotion (bottom) trials. We quantified the relationship using the slope of a linear regression fit. The four quadrants in each plot correspond to the examples in (**c**). Colorbar scale is arbitrary. **e** Subject-mean linear regression slope between signal and noise correlations, for stationary (black) and locomotion (red) trials. The triangle on the *x*-axis indicates the time point shown in (**d**). Shaded regions indicate mean ± SEM across subjects ($n = 5$). Source data are provided as a Source Data file.

despite locomotion being associated with increased firing rates, which would be expected to increase noise correlations, all other factors held constant[56,57]. As a result, signal correlations contributed more to shared population activity during locomotion (Supplementary Fig. 7).

Noise correlations peaked when firing rates decreased in stationary trials. The temporal dynamics of mean absolute noise correlations exhibited striking differences between stationary and locomotion trials. Average stationary trial noise correlations initially decreased for ~100 ms following stimulus onset, similar to locomotion trials (Fig. 3b). However, whilst noise correlations continued to decrease to a low stable level in locomotion trials, we observed transient increases in stationary trial noise correlations ~240 ms following stimulus onset and offset. These peaks in noise correlations coincided with the decreases in single-neuron firing rates we observed following stimulus onset and offset in stationary trials (Fig. 3b, dashed lines), indicating that such reductions reflect coordinated changes in neural population activity.

How do we interpret these changes in signal and noise correlations with respect to stimulus encoding? The influence of noise correlations on stimulus encoding can depend on their relationship with signal correlations[30,34,58] (but see also refs. 59, 60). Assuming homogeneous tuning curves, if pairs of neurons with high signal correlations also have high noise correlations (i.e. signal and noise correlations are themselves correlated), then these noise correlations reduce the encoding capacity of the population of neurons (Fig. 3c). If instead signal and noise correlations are anti-correlated, then noise correlations can actually increase stimulus encoding capacity (Fig. 3c). The relationship can be summarised by the slope of a linear regression fit to signal and noise correlations (Fig. 3d), whereby positive slopes indicate that noise correlations reduce stimulus encoding capacity. Reductions in the slope of signal-noise correlations have been associated with attention- and learning-based improvements in visual task performance[30,61] and have been observed with locomotion during periods of spontaneous activity[10]. We examined how the relationship between signal and noise correlations evolved over the time course of responses to understand how neural correlations affect stimulus encoding during different behavioural states.

Changes in the relationship between signal and noise correlations increased stimulus encoding capacity during locomotion. During stationary trials the slope of signal to noise correlations was positive and peaked 200–400 ms following stimulus onset (Fig. 3e), corresponding to the peak in noise correlations we observed during this period (Fig. 3b). By contrast, in locomotion trials the slope of signal to noise correlations was significantly lower ($p = 0.027$ Mixed-effects ANOVA, effect of behavioural state; $p < 0.001$ significant interaction between behavioural state and time) and close to 0 throughout. Accordingly, when we disrupted noise correlations by shuffling trials within conditions, linear decoding performance increased principally between 200–350 ms in stationary trials (Supplementary Fig. 8), the same time period that the slope of signal to noise correlations peaked (Fig. 3e). Changes in decoding performance when disrupting noise correlations during locomotion trials were in contrast small and largely time-invariant.

These results demonstrate that stimulus-evoked neural correlations dynamically evolve over the response period, and that behavioural state can strongly influence these dynamics. As a result, behavioural state can have time-varying effects on the efficiency of stimulus encoding by neural populations. In stationary trials, noise correlations dominate shared population activity (Supplementary Fig. 7) and limit information encoding capacity for visual speed, particularly in the early period following stimulus onset (Fig. 3e and Supplementary Fig. 8a, b). By contrast, during locomotion noise correlations are both reduced in magnitude and less correlated with signal correlations, resulting in increased stimulus encoding capacity and therefore more efficient sensory processing.

## Stimulus-evoked changes in correlation structure stabilise faster during locomotion

Having established how the dynamics of correlations change over time during different behavioural states, we next examined how correlations are organised. Specifically, we focused on how stable the structure of pairwise correlations was over the response period in stationary and locomotion trials. We characterised the structure of correlations using pairwise correlation matrices which provide a description of functional connectivity within a population of neurons[30,33,56]. Changes in this structure therefore indicate changes in functional connectivity. To determine how stable the structure of correlations was over time we computed Pearson's correlation between pairwise correlation matrices obtained at different timepoints (temporal correlation), separately for signal and noise correlations (Fig. 4a, b). This allowed us to quantify how stable the structure of correlations was over the response period independent of changes in their magnitude, with higher temporal correlations indicating higher stability in the structure of correlations.

The structure of signal correlations stabilised faster following stimulus onset in locomotion trials. In stationary trials the structure of signal correlations stabilised gradually over the stimulus period following stimulus onset (Fig. 4b, c). This reflected the slow emergence and strengthening of visual speed tuning during stationary trials (Fig. 2). By contrast, in locomotion trials the structure of signal correlations stabilised rapidly following stimulus onset to a higher overall level (Fig. 4b, c; $p < 0.01$ Mixed-effects ANOVA, effect of behavioural state; $p < 0.001$ significant interaction between behavioural state and time; $n = 5$ subjects). This is seen in the larger plateaus of high temporal correlation values between timepoints (Fig. 4b) and reflects the earlier and stronger visual speed tuning we observed in locomotion trials (Fig. 2). Interestingly, in locomotion trials we also observed some stability of signal correlation structure during the post-stimulus period that was distinct to signal correlation structure during the stimulus period (Fig. 4b, c), implying the presence of persistent post-stimulus tuning for visual speed that differs from stimulus-period tuning.

Stimulus-evoked changes in the structure of noise correlations also stabilised faster during locomotion. Noise correlation structure was more stable within than between stimulus and post-stimulus periods (Fig. 4b), implying that the structure of noise correlations changes between baseline and stimulus trial epochs. Since noise correlations reflect at least in part functional connectivity between neurons[30,33,56], this change in noise correlation structure indicates that changes in visual input trigger a reorganisation of functional connectivity between neurons. Indeed, stimulus onset and offset triggered a transient reduction in the stability of noise correlation structure between neighbouring time windows (Fig. 4b, c). Comparing behavioural states, we found that this stimulus-evoked reorganisation of noise correlation structure stabilised more quickly in locomotion trials, reaching maximum stability by ~200 ms, compared to ~400 ms in stationary trials (Fig. 4b, c; $p = 0.04$ Mixed-effects ANOVA, significant effect of behavioural state; $p < 0.001$ significant interaction between behavioural state and time; $n = 5$ subjects).

These findings demonstrate that changes in visual input trigger a reorganisation of functional connectivity within mouse V1, and this reorganisation stabilises faster during locomotion.

## Stimulus decoding readout stabilises faster during locomotion

How stable is stimulus information over time? Given the faster stabilisation of signal and noise correlation structure during locomotion, we hypothesised that stimulus decoding may also stabilise faster. To determine the stability of stimulus decoding between time points we tested how well decoders trained in individual time windows generalised to other time windows. We used Linear Discriminant Analysis (LDA) decoders that take into account correlations between neurons. We trained individual decoders on 100 ms time windows and tested

**Fig. 4 | The structure of pairwise correlations and stimulus decoding stabilise faster during locomotion. a** Overview of how we assessed the stability of the structure of signal and noise correlations over time. We calculated the linear correlation coefficient between pairwise correlation matrices obtained from different timepoints ('temporal correlations'). Higher correlation values indicate greater stability between two time points. Shown are pairwise signal correlation matrices from two different time points (top panels) and the linear correlation between them (bottom panel), for an example subject. **b** Matrices of the session-mean temporal correlation values (see (**a**) for calculation) for all combinations of timepoints. Shown are signal (top panels) and noise (bottom panels) temporal correlation matrices for stationary (left panels) and locomotion (middle panels) trials, as well as the difference (right panels) between them. Grey shaded regions on axes indicate the stimulus period. The dashed line in the bottom-left panel indicates neighbouring non-overlapping time windows. **c** Signal and noise temporal correlation values for neighbouring, non-overlapping 200 ms time windows. Plots are equivalent to diagonal slices as indicated by the dashed line in the bottom-left panel

of (**b**). **d** We trained and tested decoders (regularised LDA) in all combinations of non-overlapping 100 ms windows to see how well they generalised across time. Shown are matrices of session-mean decoding performance for all combinations of timepoints, for stationary and locomotion trials. **e** Comparison of session-mean decoding performance for decoders trained in a single early time window ($t = 0.1$ to 0.2; solid lines) compared to decoders that were trained and tested in each time window separately (dashed lines). The solid lines represent a horizontal slice of matrices in (**b**), indicated by the black arrows on the *y*-axis, and the dashed lines represent the main diagonal of matrices in (**b**). Decoders trained in this early time window generalised well to later stimulus periods when trained in locomotion, but not stationary trials. Inset: session-mean relative decoding performance over the stimulus period for decoders trained in this early time window ($t = 0.1$ to 0.2; S Stationary, L Locomotion). Paired two-sided *t*-test analysis ($p = 0.0137$). Shaded regions indicate mean ± SEM across subjects ($n = 5$). Source data are provided as a Source Data file.

their ability to decode visual speed from neural activity occurring in all other non-overlapping time windows (Fig. 4d). If decoding readout is stable, then a decoder trained on neural activity in one time window should be able to predict visual speed from neural activity in another time window (e.g. solid lines in Fig. 4e) as well as a decoder that was trained in that window (dashed lines in Fig. 4e), i.e. the decoder should demonstrate cross-time generalisation.

Visual speed decoding stabilised faster during locomotion. We focused our attention on time periods soon after stimulus onset, where differences in stationary and locomotion trials were most prominent. In stationary trials, decoders trained on neural activity

occurring soon after stimulus onset ($t = 0.1$–0.2 s) generalised poorly to neural activity in later stimulus periods and could only decode visual speed slightly above chance (Fig. 4e). By contrast, decoders trained during the same early time window in locomotion trials performed well predicting visual speed in later time windows. We quantified this cross-time generalisation by calculating the relative performance of decoders trained in the early time window with the performance of decoders trained and tested in each time window independently. We found a more than 2-fold increase in relative decoding performance in locomotion trials (Fig. 4e inset; Mean ± SEM stationary trials: $0.25 \pm 0.04$; locomotion trials: $0.58 \pm 0.07$; $p = 0014$ paired *t*-test;

$n$ = 5 subjects; see also Supplementary Fig. 8c). Thus, the optimal linear decoding readout of stimulus information from neural activity has already begun to stabilise 100–200 ms following stimulus onset in locomotion trials, but is still continuously evolving during this period in stationary trials.

## Neural population trajectory responses have oscillatory dynamics that are dampened during locomotion

We next examined the dynamics of neural population trajectories, which capture shared fluctuations of activity within a population of neurons and therefore provide insight into the underlying latent dynamics of a neural system[41,49]. We performed Factor Analysis (FA) on simultaneously recorded neurons to obtain latent factors that capture the dominant patterns of shared population activity (Fig. 5a and Supplementary Fig. 9). We analysed the temporal dynamics of multi-dimensional latent factor responses, which we term population trajectories. Population trajectories evolved with typical overall temporal profiles during the response period: following stimulus onset trajectories moved from a baseline steady-state to a stimulus steady-state and then returned to the baseline steady-state following stimulus offset (Supplementary Movie 1). We focused our analysis on the dynamics of population trajectories between these steady-states.

Population trajectories exhibited a range of dynamical features. Population trajectories took varied routes between baseline and stimulus steady-states (Fig. 5b). These routes could be relatively direct, with straight paths and few changes of direction, or more convoluted, with multiple changes of direction. Notably, oscillatory and spiral dynamics dominated many population trajectories. These dynamics were present across subjects and in multiple dimensions of population activity (Fig. 5b shows example pairs of latent factors from different subjects, see also Supplementary Movie 2). Given these dynamical features, we developed an analysis framework that combines established and new analysis methods to parameterise population trajectories in stationary and locomotion states.

Population trajectories make more direct transitions between steady-states during locomotion. We initially focused our analysis on the overall path taken by population trajectories during stimulus onset ($t$ = 0–0.5 s) and offset ($t$ = 1–1.5 s) trial epochs. Specifically, we assessed how direct the paths taken by population trajectories were between baseline and stimulus steady-states by calculating a ratio of the total distance travelled by a population trajectory divided by the length of the direct path between the two steady-states. We term this quantity the distance ratio (Fig. 5c). A distance ratio of 1 therefore represents the most direct trajectory possible, and larger values represent less direct trajectories. During locomotion, population trajectories took significantly more direct paths between baseline and stimulus steady states (Fig. 5d, Supplementary Fig. 10a and Supplementary Movie 2; Mean ± SEM distance ratios: stationary = 7.8 ± 0.8; locomotion = 3.2 ± 0.1; $p$ < 0.001 LME model effect of behavioural state; 5 subjects × 6 stimuli, $n$ = 30 trial-averaged responses per behavioural state). This was due to population trajectories travelling a similar distance in both behavioural states despite the stimulus steady-state being ~twice as far from the baseline state in locomotion trials (Fig. 5e; Mean ± SEM distance travelled: stationary = 22.2 ± 1.0; locomotion = 25.1 ± 1.0; $p$ = 0.22. Direct distance: stationary = 4.0 ± 0.3; locomotion = 8.1 ± 0.4; $p$ < 0.001). Population trajectories also made more direct returns to the baseline state following stimulus offset in locomotion trials (Fig. 5f, g and Supplementary Fig. 10a; Mean ± SEM distance ratios: stationary = 2.0 ± 0.1; locomotion = 1.4 ± 0.0; $p$ < 0.001). Notably, population trajectories were in general more direct following stimulus offset, indicating that features of visual inputs influence how population trajectories transition between distinct steady-states.

Population trajectories have reduced oscillatory dynamics during locomotion. We next examined the temporal dynamics of population

trajectories in more detail, focusing initially on their speed and acceleration profiles. In both stationary and locomotion trials population trajectories transiently accelerated following stimulus onset and offset, before settling to a slow trajectory speed during subsequent steady-states (Fig. 5h and Supplementary Fig. 10b). In stationary trials, stimulus-evoked changes in population trajectory speed were driven by oscillatory accelerations that lasted ~500 ms into the stimulus period. By contrast, in locomotion trials, oscillatory accelerations were dampened and population trajectories instead had a sharper speed profile with a faster maximum speed (Fig. 5h and Supplementary Fig. 10b; Mean ± SEM maximum speed: stationary = 8.5 ± 0.4; locomotion = 10.3 ± 0.4; $p$ < 0.001; LME model effect of behavioural state; 5 subjects × 6 stimuli, $n$ = 30 trial-averaged population trajectories per behavioural state). More pronounced oscillatory acceleration dynamics in stationary trials were manifested as relative spectral power being concentrated in higher frequencies (Fig. 5h inset; $p$ < 0.001 paired $t$-test).

The dampened oscillatory acceleration dynamics we observed during locomotion were associated with less oscillatory approaches to the eventual stimulus steady-state, which we assessed by parameterising population trajectories using an 'angle of approach'. Specifically, we calculated the time-varying angle between a vector that joined the position of a population trajectory with its eventual stimulus steady-state and a reference vector that joined the baseline and stimulus steady-states directly (Fig. 5i, left panel). In both stationary and locomotion trials the average angle of approach increased sharply following stimulus onset, demonstrating that population trajectories tend to deviate from the direct path between baseline and stimulus steady-states early on following stimulus onset (Fig. 5i, right panel and Supplementary Fig. 10c). Following this initial deviation, there were clear differences between behavioural states: whilst the average angle of approach gradually declined in locomotion trials, reflecting smooth, arcing population trajectories towards the stimulus steady-state, it fluctuated rapidly in stationary trials, reflecting the oscillations, spirals and frequent changes in direction taken by population trajectories in the stationary state. The more oscillatory approaches of population trajectories in stationary trials manifested as larger cumulative changes in the angle of approach towards the stimulus steady-state (Mean ± SEM cumulative change in angle: stationary = 517.0 ± 18.2; locomotion = 318.9 ± 9.5; $p$ < 0.001; LME model effect of behavioural state). We also observed a similar difference between behavioural states during the stimulus offset period (Fig. 5i, right panel and Supplementary Fig. 10d; stationary = 363.2 ± 16.0; locomotion = 220.1 ± 11.8; $p$ < 0.001).

We also found evidence of oscillatory dynamics during stationary trials directly in population mean spike counts, which manifest as increased relative spectral power between 3 ‐ 5 Hz (Supplementary Fig. 11), indicating that the oscillatory dynamics are an intrinsic property of the network during this state and not an artefact of Factor Analysis.

Collectively, these results reveal flexible neural population responses to changes in visual input, whose temporal dynamics depend on behavioural state. In stationary trials, visual stimuli evoke circuitous dynamics dominated by oscillations which eventually settle at a stimulus steady-state. By contrast, in locomotion trials neural population oscillations are strongly dampened or even absent, and population trajectories make more direct transitions between steady-states.

## Neural population dynamics are less tangled during locomotion

How robust are neural population dynamics to noise during different behavioural states? Being more robust to noise may be useful to the visual system during locomotion given that sensory inputs are likely more dynamic. We evaluated the robustness of population dynamics using neural population trajectory tangling[49], which provides insight into the underlying dynamics of neural population activity.

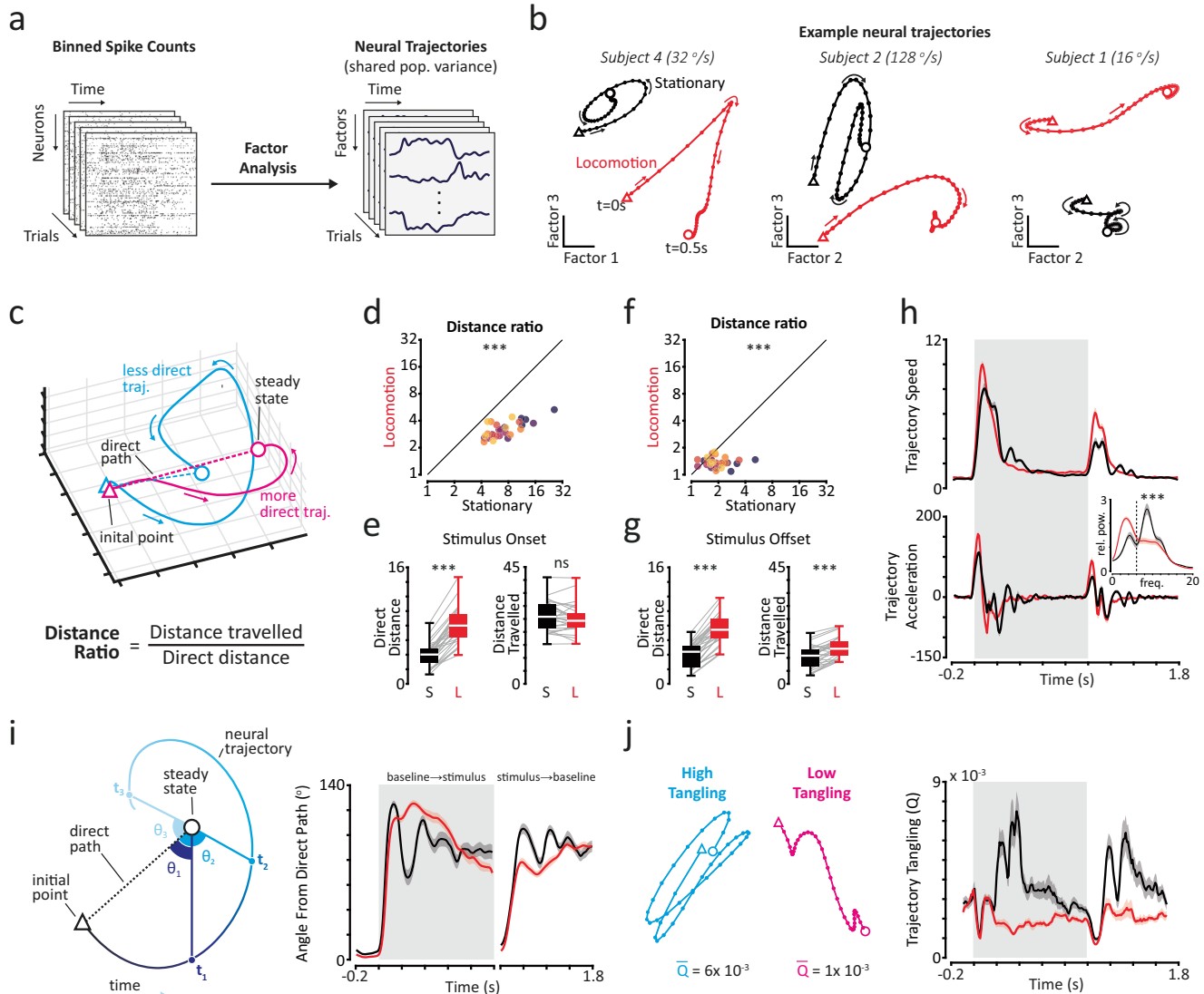

**Fig. 5 | Population trajectories are more direct, less tangled and have reduced oscillatory dynamics during locomotion. a** We performed factor analysis on smoothed, binned spike counts to obtain latent factors that represent shared population variance. **b** Example trial-averaged population trajectory responses for stationary and locomotion trials ($t = 0$ to $0.5$ s). Two factors are plotted at a time for visualisation. Arrows indicate direction of travel−trajectories start at triangles and end at circles. Dots represent 10 ms time intervals. **c** Overview of calculation of distance ratios as a measure of how direct population trajectory paths were. **d** Scatter plot of distance ratios for trial-averaged population trajectory responses during the stimulus onset period, for stationary and locomotion trials. Different colours indicate different stimulus speeds. LME analysis, two-sided effect of behavioural state. ***$p = 3.75 \times 10^{-19}$. **e** Box plots of direct distance (left) and distance travelled (right) between baseline and stimulus steady-states for trial-averaged population trajectory responses during stationary ('S') and locomotion (red, 'L') trials, for the stimulus onset period. Centre white lines indicate medians, box limits indicate upper and lower quartiles and whiskers indicate full range of data. Grey lines represent individual stimulus speeds for each subject. LME analysis

($n = 30$ paired trial-averaged responses), two-sided effect of behavioural state. ***$p = 9.30 \times 10^{-19}$; ns not significant, $p = 0.22$. **f** Same as (**d**) for the stimulus offset period. ***$p = 2.77 \times 10^{-7}$. **g** Same as (**e**) for the stimulus offset period. $p = 1.63 \times 10^{-17}$ for direct distance and $p = 2.95 \times 10^{-10}$ for distance travelled. **h** Mean trial-averaged trajectory speed (top panel) and acceleration (bottom panel) over the response period for stationary and locomotion trials. Inset: relative power spectrum of trajectory acceleration. Paired two-sided $t$-test on low frequency band (0–6 Hz). ***$p = 7.46 \times 10^{-7}$. **i** Population trajectories had less oscillatory approaches to steady-states during locomotion trials. Left panel: illustration of trajectory angle of approach analysis. Right panel: mean trajectory angle of approach over the response period for stationary and locomotion trials. **j** Population trajectories exhibited reduced neural tangling during locomotion trials. Left panel: examples of trajectories with high and low neural tangling. Mean tangling values are indicated beneath trajectories. Right panel: mean trajectory tangling over the response period for stationary (black) and locomotion (red) trials. Shaded regions indicate mean ± SEM across subjects ($n = 5$). Source data are provided as a Source Data file.

Low trajectory tangling is indicative of a dynamical system with a smooth underlying flow field and increased robustness to noise[49]. Trajectory tangling is defined as the squared difference in velocity of a population trajectory at two timepoints, divided by the squared distance between those points. Periods of high tangling thus occur when a population trajectory exhibits large changes in velocity at nearby locations such as during sharp changes in direction (Fig. 5j, left panel). Given the circuitous population trajectory dynamics we observed following

stimulus onset in stationary trials, compared to the more direct and less oscillatory dynamics in locomotion trials, we hypothesised that population trajectories would be less tangled during locomotion.

Population trajectories were less tangled during locomotion. In stationary trials, mean trajectory tangling increased following stimulus onset, peaking at ~400 ms post-stimulus onset before gradually declining to pre-stimulus baseline levels at ~800 ms (Fig. 5j, right panel and Supplementary Fig. 10e), showing that visual stimuli evoked a

prolonged period of increased population trajectory tangling. Conversely, in locomotion trials, a small increase in trajectory tangling in response to stimulus onset was followed by a rapid decline to baseline levels within ~200 ms, demonstrating that locomotion is associated with a much faster cessation of stimulus-evoked increases in population trajectory tangling ($p = 0.003$ Mixed-effects ANOVA, effect of behavioural state; $p < 0.001$ significant interaction between behavioural state and time). We also observed similar differences in trajectory tangling between behavioural states following stimulus offset (Fig. 5j, right panel), indicating that locomotion untangles population trajectory responses to changes in visual input.

### Changes in population trajectory dynamics support improved visual speed decoding during locomotion

How do changes in population trajectory dynamics affect stimulus encoding during different behavioural states? Since high trajectory tangling is associated with a reduced robustness to noise[49], we reasoned that the tangled oscillatory dynamics in stationary trials might limit the fidelity of visual stimulus encoding. We first examined how population trajectory responses varied for different visual speeds in stationary and locomotion states. Within a behavioural state, responses to different visual speeds often had similar shapes—tight spiralling trajectories in stationary trials and smooth arcing trajectories in locomotion trials—but evolved towards different eventual stimulus steady states (Fig. 6a, b and Supplementary Movie 3). Indeed, simple linear transformations could explain differences in the shapes of population trajectory responses to different stimulus speeds, but not between different behavioural states (Fig. 6c and Supplementary Fig. 12).

More divergent population trajectory dynamics supported improved visual speed decoding during locomotion. By how much and when population trajectory responses to different visual stimuli diverge is crucial to their ability to encode visual features. Population trajectories for different visual speeds quickly began to diverge following stimulus onset in both stationary and locomotion trials (Fig. 6a, b, d). In stationary trials, population trajectories for different speeds reached their maximum divergence at ~100 ms, the same time point that they began to exhibit oscillatory dynamics. In locomotion trials, population trajectories continued to diverge after this time point as they carried on more arcing paths (Fig. 6d and Supplementary Movie 3). Overall, population trajectory responses to different visual speeds diverged more than twice as far from each other in locomotion trials (Mean ± SEM distance between trajectories over the stimulus period: stationary = 2.87 ± 0.49; locomotion = 5.95 ± 0.54; $p < 0.001$ paired $t$-test; $n = 5$ subjects). When we decoded visual speed from population trajectories we confirmed that decoding performance increased slowly following stimulus onset in stationary trials (Fig. 6e), and continued to increase even after trajectories for different visual speeds had maximally diverged (Fig. 6d), but while trajectory tangling continued to decrease (Fig. 5j). By contrast, decoding performance in locomotion trials increased rapidly to a high stable level (Fig. 6e), reflecting the increased trajectory divergence (Fig. 6d) and rapid reduction in trajectory tangling (Fig. 5j). Notably, the largest improvement in decoding performance was during the initial period of responses following stimulus onset (Fig. 6e; $p < 0.01$ Mixed-effects ANOVA, effect of behavioural state; $p < 0.001$ significant interaction between behavioural state and time), when population trajectories moved with oscillatory, tangled dynamics in stationary trials compared to the smooth, stable, arcing population trajectories present in locomotion trials (Fig. 6a, b).

These findings reveal how behavioural state and stimulus features differentially modify population trajectory response dynamics. Whilst changes in visual speed are associated with simple linear transformations, behavioural state has a more profound effect, not easily captured by simple linear transformations. During locomotion, visual speed-dependent transformations of population trajectory dynamics result in stimulus-evoked steady states that are more spread out, which alongside less tangled neural population dynamics, improves population encoding of visual speed. Additionally, when we decoded visual speed from latent factors, improvements in performance during locomotion were most pronounced in the early time period following stimulus onset, potentially facilitating the rapid encoding of behaviourally-relevant visual inputs during this state.

## Discussion

Using large-scale electrophysiology[51] to simultaneously record from hundreds of neurons in the primary visual cortex of awake, behaving mice, we have shown that sensory systems can adapt during different behavioural states by modifying the temporal dynamics of neural responses. We observed changes in the temporal dynamics of single-neuron firing rate responses (Figs. 1 and 2), the magnitude and structure of pairwise neural correlations (Figs. 3 and 4) and trajectories of population activity (Figs. 5 and 6) between inactive, stationary behavioural states and active states characterised by locomotion. Functionally, visual speed decoding from neural population activity in stationary states was initially weak and unstable following stimulus onset, improving only slowly over time. Rapid motion perception of new visual inputs may therefore be poor in this behavioural state. In contrast, changes in temporal population dynamics during locomotion mediated rapidly stabilising and accurate visual speed decoding, reflecting the increased importance of quickly perceiving and responding to new visual motion inputs during active movement through an environment.

Stability of sensory coding at sub-second timescales is influenced by behavioural state. During inactive, stationary states, stimulus onset triggered a period of reduced stability in neural population dynamics: single-neuron firing rates transiently increased[2,4], changes in the structure of pairwise correlations stabilised slowly and population trajectories exhibited tangled[49], oscillatory dynamics which took circuitous routes between baseline and stimulus encoding neural states. By contrast, during locomotion, single-neurons had less transient response dynamics, stimulus-evoked population activity rapidly reached a stable, decorrelated state and population trajectories transitioned more directly, with less oscillations, between baseline and stimulus-encoding neural states. These findings suggest that new visual information, as occurs following stimulus onset, is more stably integrated into ongoing neural population activity during locomotion. Indeed, the decoding readout of visual information from V1 population activity stabilised rapidly following stimulus-onset during locomotion, but continuously evolved during stationary states (Fig. 4d, e). This stabilised sensory representation in V1 may facilitate the transmission of visual information to downstream areas during active behavioural states.

Oscillatory dynamics dominate neural population trajectory responses to visual stimuli during stationary states, but are dampened during locomotion. Previous research has identified the presence of oscillations in motor cortex neural population dynamics, even in the absence of rhythmic movements[62]. The moving dot field stimuli we used in these experiments similarly lack overt oscillatory properties, suggesting that the oscillatory neural population dynamics we observed are a result of intrinsic network properties. Moreover, neural population oscillations were independent of visual stimulus speed and contingent on behavioural state, further ruling them out as a trivial result of visual stimulation. Similarly to primate motor cortex[62,63], single-neuron responses in mouse V1 were heterogeneous and often had multiphasic dynamics in stationary trials (Fig. 1c–e). Yet the coordinated neural population oscillations we observed are not readily inferred from these single-neuron responses. Indeed, oscillatory dynamics were present in many latent factors, including those that represented the strongest modes of shared population activity

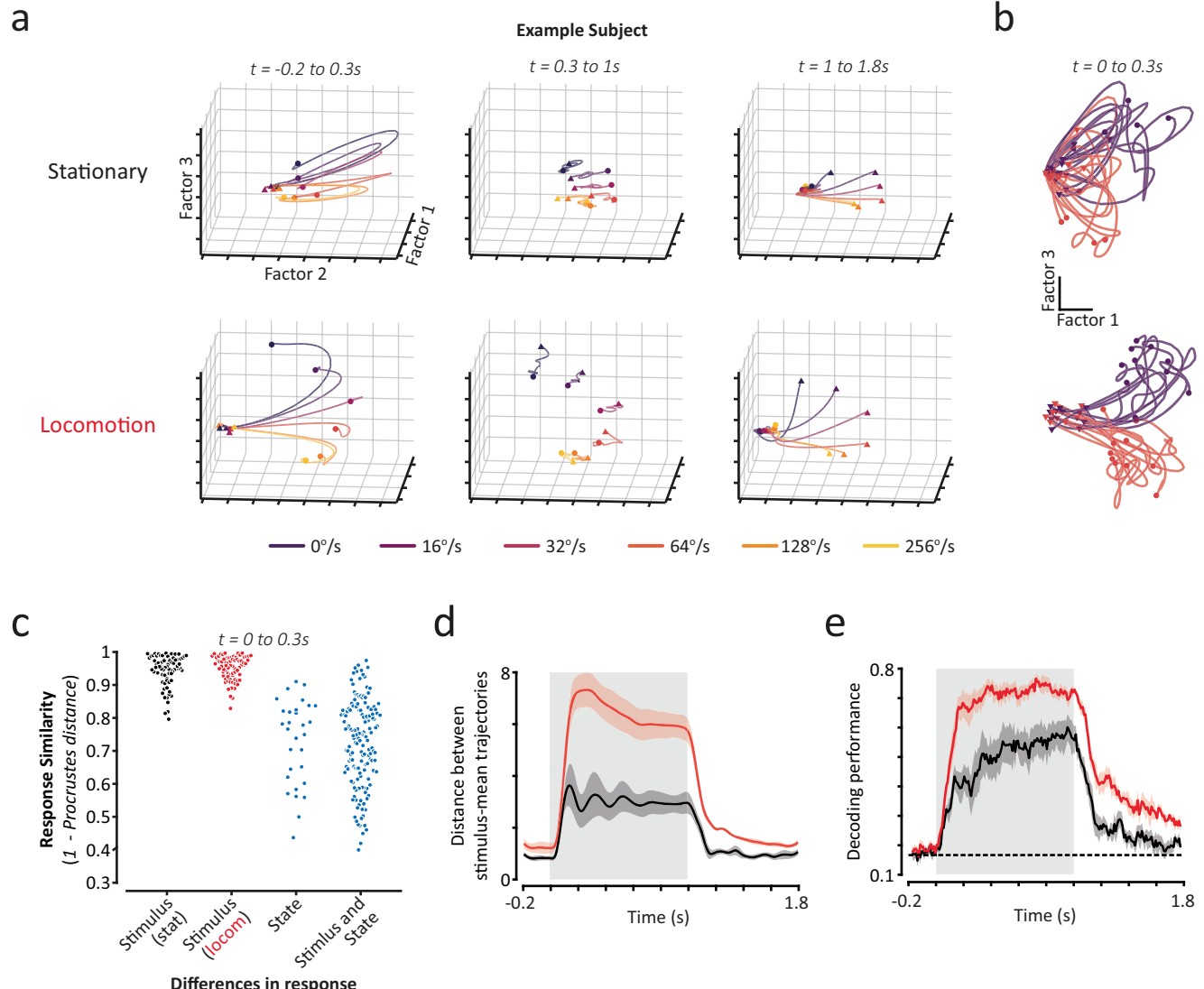

Fig. 6 | **More divergent population trajectories support improved visual speed decoding during locomotion. a** Trial-averaged population trajectory responses to all stimulus speeds presented, for an example subject. Population trajectories for stationary (top panels) and locomotion (bottom panels) trials are shown, split by trial epoch. The first three latent factors (representing the most shared variance) are plotted for visualisation. Triangles and circles mark the start and end of a trajectory. **b** Example single-trial trajectories from stationary (top panel) and locomotion (bottom panel) trials (10 randomly selected trials for 16°/s and 64°/s) for the same example subject as in (**a**). Two latent factors are plotted for visualisation. **c** Procrustes similarity (rigid shape analysis) of population trajectory responses ($t = 0$ to $0.3$ s) to different stimulus speeds ('Stimulus') and during different behavioural states ('State'). Each data point represents a pair of trial-averaged responses from a subject. **d** Mean distance between trial-averaged trajectory responses to the six stimulus speeds presented, for stationary (black) and locomotion (red) trials. **e** Subject-mean decoding performance (regularised LDA) over the response period (50 ms sliding window; 10 ms step size) using population trajectory responses from stationary (black) and locomotion (red) trials. Shaded regions indicate mean ± SEM across subjects ($n = 5$). Source data are provided as a Source Data file.

(Fig. 5b), demonstrating that they are a dominant feature of neural population responses in mouse V1 during stationary states.

Increased oscillatory dynamics were also apparent in population mean spike counts during stationary trials in the form of increased relative spectral power between 3 ‑ 5 Hz (Supplementary Fig. 11). Previous analyses of local field potentials (LFP) and membrane potentials in mouse V1 have shown that 3 ‑ 5 Hz power increases are associated with reduced visual task performance or task disengagement[18,64,65], as well as potentially immobility and low arousal more generally[66] (but see ref. 64). This is in agreement with our findings that latent population trajectory oscillations are associated with poorer neural encoding of visual speed and occur during stationary states. The damping of these population oscillations during locomotion reflects to some extent the less transient and more sustained stimulus-evoked

firing rates we observed in this state. These findings establish that behavioural state has a profound influence on the latent dynamics of neural population responses in mouse V1.

Changes in functional connectivity may mediate state-dependent neural population dynamics. Responses from stationary and locomotion trials differed most strikingly between 100 ‑ 500 ms following stimulus onset, a period of time where top-down interactions may dominate[43]. Top-down and recurrent functional connectivity has been hypothesised to be strongly dependent on behavioural state[27,39,40] and to play a prominent role in the temporal dynamics of neural activity following the initial stimulus onset response[2,3,67]. An array of visual and non-visual areas convey diverse top-down neural signals[17,40,68-73], which can alter functional connectivity within V1[27,36-39,74], consistent with the changes in neural correlations between stationary and locomotion

states that we observed. Faster stabilisation of neural correlations and reduced trajectory tangling during locomotion suggests that these changes in functional connectivity strengthen intrinsic neural population dynamics to enable more stable and noise-robust neural responses to new visual input during this state. To what extent reduced trajectory tangling during locomotion in mouse V1 is related to untangled population activity in mouse motor cortex remains an open question[49]. Future research should be able to disentangle how between-area interactions shape neural population dynamics during different behavioural states using experiments combining multi-area recordings and population-level analyses[42,43,75,76], alongside spatially and temporally targeted perturbations of neural activity[40,77].

What is the function of transient onset responses in stationary states? The peak in noise correlations following stimulus onset in stationary trials was aligned with signal correlations, increasing the redundancy of population coding for visual speed[30]. Our analysis of single-neuron responses from the Allen Institute's 'Visual Coding' dataset[55] (Supplementary Fig. 3), as well as previous findings[31], suggest that this early redundant population response may be widespread in the mouse visual system, and is associated with a peak in inter-area fluctuations[31]. It may function to signal only limited features of visual input[4,31] (for example detecting a change in visual input rather than real-time stimulus feature coding), perhaps to economise sustained stimulus-evoked firing rates, whilst locomotion accelerates the real-time encoding of visual motion at the expense of elevated firing rates.

The decorrelation of population responses has been proposed to contribute to improved sensory encoding during locomotion and active behavioural states[7,10,11,18]. Our results suggest that the disruptive effects of noise correlations on stimulus encoding during stationary states are time-dependent and occur primarily 200 - 400 ms following stimulus onset in mouse V1 (Fig. 3e and see also Supplementary Fig. 8a, b). Notably, the benefits of removing noise correlations (by shuffling trials) on linear decoding of visual speed performance scaled with population size in both stationary and locomotion trials, indicating that information-limiting correlations are present in both behavioural states (Supplementary Fig. 8a, b)[59,60,78]. Precisely how behavioural state influences the magnitude and temporal dynamics of information-limiting correlations with respect to sensory encoding remains an important open area of research.

Changes in noise correlations between behavioural states cannot be explained by non-specific secondary factors. While noise correlations are known to be modulated by many factors[56,57], our results cannot be explained by them. Firstly, whilst noise correlations generally increase with firing rates, we observed a decrease in noise correlations during locomotion, when firing rates were higher. Moreover, noise correlations exhibited a pronounced peak at ~240 ms following stimulus onset in stationary trials, concurrent with decreases in firing rates from their initial transient peaks. Secondly, neurons in mouse V1 are tuned to locomotion speed[79] and covarying responses to changes in locomotion speed should also increase the magnitude of noise correlations during locomotion. The reduction in noise correlations during locomotion can therefore be interpreted as being despite these two secondary factors. Finally, spike-sorting errors are known to affect estimates of noise correlations, but we compared the same neuronal pairs in stationary and locomotion states, precluding this from biasing our results.

Our findings are consistent with a range of inputs to mouse V1 contributing to changes in temporal dynamics. Neurons in mouse V1 have depolarised and less variable resting membrane potentials during locomotion, which depends on neuromodulatory and thalamic inputs[12,53,66]. Altered intrinsic membrane properties could conceivably contribute to the state-dependent firing rate and emergent population dynamics we observed here. Thalamic neurons transition from bursty to tonic firing modes during locomotion[10,15,66], akin to the transition from transient to sustained response dynamics we observed in mouse V1. Moreover, thalamic inputs are necessary for subthreshold membrane potential oscillations in V1 neurons[66], indicating that thalamic inputs may play a key role in modifying temporal response dynamics in mouse V1[80]. In stationary trials, noise correlations exhibited a pronounced peak at ~240 ms following stimulus onset, alongside the suppression of firing rates from their initial transient peaks. Since noise correlations reflect shared functional connectivity[30,33–35,56], this peak in noise correlations may be a manifestation of a suppressive feedback input into mouse V1 that regulates V1 firing rates. Multiple neuromodulatory systems innervate mouse visual cortex and display co-ordinated changes in activity during arousal and locomotion[81,82], contributing to a number of state-dependent changes in visual cortical activity[17,37,72,73,83]. In particular, cholinergic stimulation reduces noise correlations and increases encoding efficiency in mouse V1 by reducing the alignment of noise correlations with signal correlations[84], as we observed during locomotion (see also ref. 72). A complex interplay of factors are therefore likely to contribute to state-dependent temporal dynamics in mouse V1.

To what extent can we disentangle the effects of different aspects of behavioural state? The multitude of mechanisms underlying state-dependent modulation of neural activity reflect that behavioural states consist of a complex mixture of self-motion and cognitive state variables. Dissociating their effects is a key challenge in behavioural neuroscience[6,11,21,85,86]. Here we leveraged spontaneous locomotion behaviour of mice to compare neural responses during inactive, stationary states and active states characterised by locomotion. Locomotion is also associated with a number of cognitive state changes including increased arousal as indexed by pupil dilation[10,21], which can have distinct effects on mouse V1 activity[11,21]. The potentially intermediate effects of heightened pupil-indexed arousal when compared to locomotion (Supplementary Fig. 5) suggest that the changes in temporal response dynamics may link to arousal or activity levels more generally, and are not necessarily specific to locomotion. Future experiments combining detailed self-motion and physiological measurements[46,86–90] alongside neural recordings and active perceptual tasks will help elucidate how the various factors associated with behavioural states influence the temporal dynamics of neural responses and sensory perception[6].

How might response dynamics flexibly change in other brain areas with behavioural states? Our analysis of Allen Institute's 'Visual Coding' dataset[55] revealed comparable state-dependent changes in the response dynamics of single-neurons in mouse higher visual cortical areas to those we observed in primary visual cortex. This suggests coordinated behavioural-state related changes in population response dynamics across all visual areas. In somatosensory cortex locomotion is associated with depolarised and less variable membrane potentials, increased amplitude of touch stimulus-evoked single-neuron responses and reduces pairwise noise correlations[19,91,92], similar to primary visual cortex. Touch stimuli appear to evoke both transient and sustained response dynamics in mouse somatosensory cortex[92], and it would be interesting to investigate their state-dependent temporal dynamics at high temporal resolution. Whether flexible state-dependent response dynamics such as those we have observed in mouse visual cortex are also present in brain areas that encode different sensory modalities remains an important open question.

In summary, our findings establish a principle of how sensory systems adapt to changing perceptual demands, such as during different behavioural states, where flexible neural population dynamics govern the speed and stability of sensory encoding. Our results provide important constraints on modelling real-time processing of sensory inputs. Moreover, whilst we have developed an analysis framework to study temporal dynamics of visual processing in mouse V1, our methods are readily applicable to other brain areas, sensory modalities and species[93,94]. The broader application of this framework should help to uncover further principles of sensory processing at subsecond timescales.

## Methods
### Data collection
All experiments were performed in accordance with the Animals (Scientific Procedures) Act 1986 (United Kingdom) and Home Office (United Kingdom) approved project and personal licences. We additionally analysed in vivo extracellular electrophysiology recordings of eight mouse visual areas using the 'Visual Coding' dataset from the Allen Institute for Brain Science[55].

**Experimental subjects.** In recordings that we conducted, we used C57BL/6J wild-type mice ($n = 5$; all female, age 9–18 weeks during recordings) obtained at approximately 7 weeks of age from Charles River UK Ltd. Mice were individually housed under a reversed 12-h light/dark cycle and experiments were performed during the dark phase of the cycle. The temperature ranged from 19–23 °C and the humidity was 55% ± 10%. Mice had free access to food and water. Sex was not considered in our study design or analysis as it was unlikely to be relevant to our scientific findings. Additionally we confirmed a subset of our results in the Allen Institute for Brain Science dataset which contains both sexes.

The Allen Institute for Brain Science dataset included recordings from 24 mice ($n = 8$ female, $n = 16$ male; $n = 12$ wild-type C57BL/6J, $n = 3$ Pvalb-IRES-Cre × Ai32, $n = 6$ Sst-IRES-Cre × Ai32, $n = 3$ Vip-IRES-Cre × Ai32; age 15–20 weeks during recordings). These mice were kept in a reversed 12 h light cycle between 20–22 °C at 30–70% humidity. Experiments were performed during the dark phase of the light cycle.

**Surgeries.** We implanted mice with a custom stainless-steel headplate attached to the skull to enable head-fixation. Surgeries were performed under isoflurane anaesthesia (induced at 3% and maintained at ~1.5%) and mice were allowed to recover for 7 days with analgesia for 3 days before habituation sessions began. Mice were then progressively habituated to the experimental apparatus for longer time periods (5–30 min) until they were perceived as comfortable locomoting on the polystyrene wheel (6–8 sessions). Following habituation, a craniotomy over V1 (AP = −3.5; ML = +2.5) was performed under anaesthesia. We covered the exposed dura mater with Dura-Gel (Cambridge NeuroTech) and then placed a protective plastic cap over the craniotomy which we sealed with a silicone elastomer (Kwik-Cast, WPI). Mice were allowed to recover for at least 20 h before recordings began.

**Visual stimulation.** Visual stimuli were projected onto the interior surface of a truncated dome[52] (60 Hz frame rate). We performed a custom mesh-mapping procedure using Bonsai software[95] to map projector-based pixel coordinates to a visual angle-based coordinate system. We gamma-corrected the display using standard methods. During experiments, mice were head-fixed in the geometric centre of the dome on top of a polystyrene wheel which they were free to locomote on. The full display surface spanned −120° to 120° azimuth and −30° to 90° elevation (where 0° is the horizon) of visual angle. Stimuli were presented in the contralateral visual field to the recorded brain hemisphere and spanned −120° to 0° azimuth and −30° to 80° elevation.

Visual stimulation was designed and controlled with customscripts using BonVision[96], an open-source package for visual environment generation within Bonsai[95] software. Visual stimulation consisted of a series of trials (1 s duration) separated by a mid-grey interstimulus-interval (1 s duration). Each trial consisted of a field of randomly positioned full contrast and full opacity black and white circles (2° diameter, 12.5% max density—effective density was lower due to occlusion) over a mean luminance background. These parameters were chosen based on previous studies[8,55] and our own experiments showing that they evoke strong responses in mouse V1. We layered black and white circles such that they occluded each other approximately equally. On each trial all circles moved in the naso-temporal direction at the same visual speed, which varied randomly between trials. We chose to vary visual speed as we believe it to be a highly behaviourally relevant variable during locomotion. The six visual speeds presented were 0, 16, 32, 64, 128 and 256°/s. These speeds were chosen based on a preliminary analysis of the 'Visual Coding' dataset from the Allen Institute for Brain Science[55] which showed that they evoke strong responses in mouse V1. Each visual speed was presented between 100–200 times within a session (mean = 155 trial repeats). A small quad which flipped between black and white was also presented in the inferior/peripheral part of the ipsilateral visual field to generate a photodiode signal that enabled us to precisely measure the frame presentation times at which stimulus presentation started and finished.

**Electrophysiology recordings.** We performed acute, in vivo electrophysiology recordings using 4-shank neuropixel 2.0 probes[51] (Imec) to record from mouse primary visual cortex. In one mouse we used a single-shank 1.0 neuropixel probe[97]. Data were acquired via a PXI system (National Instruments) using SpikeGLX software. At the start of each session a protective plastic cap sealed with a silicone elastomer (Kwik-Cast, WPI) was removed to reveal the craniotomy site. We then slowly lowered the probe using a micromanipulator (uMp-4, Sensapex) until it was positioned such that all active recording banks (2 banks closest to the tips on each shank) were within V1. We left the probe to settle within the brain for ~10 min before recording. Only one recording session was performed per subject for this stimulus set.

Alongside electrophysiology data we also recorded wheel movement using a rotary encoder (05.2400.1122.1024, Kübler) to calculate locomotion speed. We also captured video recordings of the face of the mouse using a camera (DMK 27BUR0135, The Imaging Source) with a zoom lens (MLH10X, Computar) to measure pupil dilation. Electrophysiology, photodiode and behavioural signals were all recorded alongside an asynchronous digital pulse generated by an Arduino Leonardo (Arduino) microcontroller to allow for post hoc synchronisation of data streams.

**Allen Institute dataset.** We analysed sessions where moving dot field stimuli were presented (*Functional Connectivity* stimulus-set). These recordings simultaneously targeted eight mouse visual areas (Cortical areas: primary visual cortex (V1), lateromedial cortex (LM), anterolateral cortex (AL), anteromedial cortex (AM), posteromedial cortex (PM). Thalamic areas: dorsal lateral geniculate nucleus (dLGN), lateral posterior nucleus (LP)) using six Neuropixel probes[97]. Dot fields consisted of ~200 3° diameter white dots moving across a mean-luminance grey background. In a given trial all the dots moved at one of seven visual speeds (0, 16, 32, 64, 128, 256, 512°/s) and in one of four directions (−45°, 0°, 45°, 90°; where 0° is nasal to temporal motion and positive changes indicate clockwise rotation) at 90% coherence. Stimuli were repeated 15 times in a random order.

For each session and stimulus condition we assessed whether at least 10/15 trials were viewed whilst mice were in a single behavioural state (stationary or locomotion). We used the same criteria to classify trials as stationary and locomotion as we did for our own recordings. Where sufficient trials were available for a given behavioural state, we analysed those trials and discarded the remaining trials. Using this criteria, we analysed 56,646 responses recorded in stationary trials (3447 cells, 15 sessions) and 43,767 responses recorded in locomotion trials (3031 cells, 12 sessions).

## Data analysis
### Behavioural analysis
**Locomotion speed.** Locomotion speed was calculated by first converting rotary encoder ticks to linear distance based on the radius of the wheel and then taking the temporal derivative of this distance

between time bins (60 Hz sampling rate). We then resampled wheel speed into 10 ms time bins and smoothed this vector using a gaussian kernel with a 35 ms standard deviation.

**Pupil dilation.** Pupil dilation was estimated from video frames using DeepLabCut[98]. We trained a network (ResNet-50) to predict 8 equally spaced points around the perimeter of the pupil using manually labelled frames. For each video frame, we then used these points to fit an ellipse (Matlab function *ellipticalFit*) to the pupil and calculated the area of the ellipse as an estimate of pupil dilation. We then resampled pupil dilation to 10 ms time bins and smoothed this vector using a gaussian kernel with a 35 ms standard deviation.

**Trial classification according to behavioural state.** We classified trials as 'stationary' and 'locomotion' based on locomotion speed. For each trial we considered the time window from 200 ms before stimulus onset to 800 ms after stimulus offset (2000 ms total). Trials were classified as stationary if mean locomotion speed during this time window was <0.5 cm/s and locomotion speed was <3 cm/s for >75% of the duration of the window. Trials were classified as locomotion if mean locomotion speed was >3 cm/s and locomotion speed >0.5 cm/s for >75% of the window. Trials that did not meet either of these criteria were excluded from analysis. In each session we downsampled trial counts such that each condition (6 stimuli × 2 behavioural states = 12 conditions) had an equal number of trials.

We classified stationary trials as low arousal or high arousal based on pupil dilation to determine the effects of arousal on temporal dynamics in the absence of locomotion. In each session we first calculated the mean pupil dilation during the stimulus period for each trial and then found the tertiles of this distribution. Trials with pupil dilation lower than the first tertile were classified as low arousal and trials with pupil dilation higher than the second tertile were classified as high arousal. Trials with pupil dilation between these two tertiles were not included in this analysis.

We classified locomotion trials as having 'slow' and 'fast' locomotion speeds to determine the effects of locomotion speed on single-neuron temporal response dynamics. To partition trials, we found the quartiles of mean stimulus period locomotion speed independently for each subject and classified 'slow' trials as trials with the slowest 25% of locomotion speeds and 'fast' trials as trials with the fastest 25% of locomotion speeds, in order to maximise the difference between the two conditions.

We also tested whether our results were robust to two alternative behavioural state criteria for classifying stationary and locomotion trials. One criteria, 'stricter', was a stricter version of our main criteria where trials were classified as stationary if wheel speed was <0.5 cm/s throughout the 2 s trial period and as locomotion if mean wheel speed was >3 cm/s and wheel speed was >0.5 cm/s for ≥90% of 2 s trial period. The other criteria, 'changepoints', used a changepoints analysis to classify epochs of continuous locomotion[99]. First, unprocessed wheel speed was z-scored and then smoothed using a 400 ms gaussian kernel. Initial locomotion epoch onset and offset times were then defined as the point at which this processed wheel speed exceeded and fell under a threshold of 0.05. We then excluded epochs with a mean locomotion speed <3 cm/s or duration <5 s. We next added 0.5 s to each locomotion epoch onset time and subtracted 0.5 s from each locomotion offset time, adding an additional buffer to ensure we did not include trials in which locomotion had recently begun or was about to finish, respectively. We then classified trials as locomotion trials if they started and finished within each epoch (i.e. the full 2 s trial period from 200 ms pre-stimulus onset to 800 ms post-stimulus offset was contained within a valid locomotion epoch). Reducing the 'window length' of the analysis from 2 s to shorter lengths and altering the z-score threshold did not have a notable effect on classification of locomotion epochs, so we opted to follow the parameters set out in ref. 99.

**Electrophysiology post-processing and spike-sorting.** We used CatGT (SpikeGLX) to perform a modified common average referencing (global demux filter option) and align electrophysiology with asynchronous pulse times for synchronisation with other data streams.

We used Kilosort 3[100] to spike-sort electrophysiology data. We then ran a series of post-processing modules[55] to remove putative double-counted spikes, label noise clusters, calculate mean spike waveforms for each cluster and generate a series of quality metrics for each cluster. We only analysed clusters which we classified as 'good' based on 3 criteria[101]: (1) Refractory period violations ≤10%; (2) Amplitude distribution cut-off ≤10%; and (3) Mean amplitude ≥50 uV.

**Cell-type classification.** We classified good clusters as three different cell-types (pyramidal, narrow interneuron, wide interneuron) based on their electrophysiological properties[54]. Specifically, clusters were classified based on the burstiness of spiking activity (measured using the rise time of a triple exponential fit to autocorrelograms (τ), and waveform duration (trough-to-peak time). Clusters were classified as narrow interneurons if their waveform duration was ≤450 μs; as pyramidal neurons if their waveform duration was >450 μs and $\tau \leq 6$ ms; and as wide interneurons if their waveform duration was >450 μs and $\tau > 6$ ms.

## PSTH responses

To obtain trial-averaged response profiles of single-neurons as a function of time we constructed peri-stimulus time histograms (PSTHs). We constructed PSTHs for each cell, stimulus speed, and behavioural state separately. We first calculated spike counts in 10 ms time bins from 200 ms pre-stimulus onset to 800 ms post-stimulus offset (2000 ms total). We then took the mean responses across trials and smoothed the resultant binned average response using a Gaussian kernel with 35 ms standard deviation.

**PSTH reliability.** We determined if PSTHs were reliable using three criteria: (1) the firing rate range was ≥3 Hz. (2) the maximum absolute value of the z-scored PSTH was ≥3.29 (i.e. 99.9% confidence interval). Z-scored PSTHs were calculated based on the mean and standard deviation of firing rates during blank trials where no stimulus was shown. (3) The PSTH passed a cross-validation reliability threshold. For each cross-validation iteration, we randomly split trials into two equal sets and constructed two PSTHs (from stimulus onset to 800 ms post stimulus offset (1800 ms total)). We then calculated the linear correlation between the time-binned firing rates (spikes/s) of the two PSTHs. Additionally, for each cross-validation iteration we also randomly shuffled the spike times in one set of trials and constructed a PSTH as usual from these shuffled spike times. We then calculated the linear correlation between the normally constructed PSTH (from one half split of trials) and the shuffled PSTH (from the other half split of trials). For each cross-validation iteration, we repeated this shuffling procedure 40 times. We performed 25 cross-validation iterations to obtain a distribution of 25 normal correlation estimates and 1000 shuffled correlation estimates. We then calculated the mean and standard deviation of the shuffled correlation distribution and used this to calculate a z-score for the mean of the normal correlation distribution. Finally, we determined a PSTH response to have a reliable shape if this z-score value was ≤−1.645 (equivalent to $p \leq 0.05$).

To test if there was a significant difference in the proportion of responses that were classified as reliable in stationary and locomotion trials we performed a McNemar test.

**PSTH sorting.** We sorted PSTH responses that were reliable in both stationary and locomotion trials using a hierarchical clustering procedure[102]. We performed the sorting algorithm on responses from stationary trials and applied the resultant ordering on responses from locomotion trials. First, we normalised the firing rates of PSTHs to between 0 and 1. We then computed a dissimilarity matrix X(N × N)

(N = number of reliable responses) where X(i,j) is the distance between PSTHi and PSTHj. As a distance measure we used Dynamic Time Warping (DTW) with a maximum stretch value of 10 time bins (100 ms) and the time bin-summed Euclidean distance. DTW enables the alignment of time series with similar shapes which are otherwise not precisely aligned in time. Having obtained the dissimilarity matrix, we performed hierarchical clustering using the unweighted average distance (Matlab function *linkage*). We then obtained the optimal ordering of PSTHs using an algorithm that minimises the sum of pairwise distances between neighbouring leaves of a dendrogram[103] (Matlab function *optimalleaforder*).

**PSTH onset and offset feature classification.** In order to classify the onset and offset features of PSTHs we first normalised the firing rates of PSTHs to between 0 and 1 and extracted the onset period (0–0.3 s, where 0 s was stimulus onset) and offset period (1–1.5 s, where 1 s was stimulus offset) of each normalised PSTH. We then separately fit a series of descriptive functions (*Decay*, *Rise*, *Peak*, *Trough* and *Flat*) to the onset and offset periods of each PSTH using the Matlab function *lsqcurvefit*.

Descriptive function were Gaussian of the form:

$$f(x) = b + ae^{\frac{(x-u)^2}{2\sigma^2}} \tag{1}$$

where $b$ is a baseline firing rate parameter, $a$ is an amplitude parameter, $u$ is the mean of the distribution and $\sigma$ the standard deviation. $x$ represents the zero-based time bins the functions were fit over.

Different parameter bounds were used to differentiate the descriptive functions. *Decay* and *Rise* features were described by two functions each (with positive and negative amplitudes). *Decay*, *Rise*, *Peak* and *Trough* features had appropriately bounded means ($u$). The upper bound for the standard deviation ($\sigma$) was lower for *Peak* and *Trough* features to ensure well defined maxima or minima. *Trough* was differentiated from *Peak* by enforcing a negative amplitude ($a$). *Flat* was primarily defined by enforcing a small amplitude parameter ($a$).

Additionally, *Peak* and *Trough* features required the signed peak of the fitted function to have a prominence ≥0.2 (calculated using the Matlab function *findpeaks*). Moreover if the range of the normalised response over the onset or offset period was <0.2 (i.e. only a small amount of firing rate modulation) the feature was classified as *Flat* by default.

To test if there was a significant difference in the proportion of reliable responses with a given onset or offset feature between behavioural state we used binomial generalised linear mixed effects (GLME) models. We fit a model for each onset and offset feature independently. The models had the formula:

$$feature \sim 1 + state + (1|subject) \tag{2}$$

where *feature* is a binomial response variable coding for whether a response had (1) or didn't have (0) a specific onset/offset feature, 1 is a fixed intercept term, state is a fixed effects term coding for whether the response was obtained from stationary or locomotion trials, (1|*subject*) is a random intercept grouped by subject.

For our analysis of responses to different visual speeds, responses of different cell-types and for responses from cells in different visual areas we fit separate models for each speed, cell-type and visual area.

**Sustainedness index.** To determine how sustained PSTH responses were, we calculated a sustainedness index based on a previously used metric[104]:

$$sustainedness\ index = mean_{\Delta FR} \div peak_{\Delta FR} \tag{3}$$

where $mean_{\Delta FR}$ and $peak_{\Delta FR}$ are baseline corrected mean and peak firing rates, respectively. To calculate $mean_{\Delta FR}$ and $peak_{\Delta FR}$ we first determined whether the mean firing rate of the PSTH was greater than or less than the baseline rate obtained from blank trials where no stimulus was presented. If the mean firing rate was greater than baseline rate we calculated $mean_{\Delta FR}$ and $peak_{\Delta FR}$ by subtracting the baseline firing rate from the mean and peak firing rate. If the mean firing rate was less than the baseline rate (i.e. the response was suppressed overall), we calculated $mean_{\Delta FR}$ and $peak_{\Delta FR}$ by subtracting the mean and peak firing rate from the baseline rate. Using this approach allowed us to determine how sustained a response was regardless of whether it was excitatory or suppressed.

To determine if the sustainedness of responses was significantly different between behavioural states we used a linear mixed-effects (LME) model with the following formula:

$$values \sim 1 + state + (1|cell) + (1|subject) \tag{4}$$

where *values* is the response variable containing sustainedness index values, 1 is a fixed intercept term, state is the fixed effect term coding for whether a response was obtained from stationary (0) or locomotion (1) trials, (1|*cell*) and (1|*subject*) terms are random intercepts grouped by cell and subject.

## Visual speed tuning

We assessed the tuning strength of individual neurons over time using the cross-validated Coefficient of Determination ($R^2$). For each 200 ms sliding window (10 ms step size) we performed 3-fold cross-validation where 2/3 of trials were randomly sampled as a training set and the remaining 1/3 as a test set. For each iteration of cross-validation we constructed two models using the training set: a tuning curve model (*trained model*) based on the mean spike count response to each visual speed and a *null model* calculated as the mean spike count to all visual speeds combined. Using the test set we constructed a mean spike-count tuning curve model (*test model*) in the same way as the *trained model*. We then determined how well the *trained model* and *null model* could predict the *test model* by calculating the sum-of-squared residuals between them. The coefficient of determination was then calculated with the following equation:

$$R^2 = 1 - \frac{SS_{model}}{SS_{null}} \Big\} \ if\ SS_{model} \leq SS_{null}$$
$$R^2 = -1 + \frac{SS_{null}}{SS_{model}} \Big\} \ if\ SS_{model} > SS_{null} \tag{5}$$

where $SS_{model}$ is the sum of squared residuals between the *trained model* and the *test model* and $SS_{null}$ is the sum of squared residuals between the *null model* and the *test model*.

We then computed the mean $R^2$ value over the 3 cross-validations, using a unique set of test trials on each iteration. We repeated this process 10 times with different random splits of train and test trials to obtain 10 estimates of $R^2$. Our final estimate of $R^2$ was taken as the mean of these 10 values. We obtained a shuffled distribution of $R^2$ values for each neuron by performing the same 3-fold cross-validation procedure on randomly shuffled spike counts 100 times.

We considered a neuron to be tuned during a given time interval if $R^2 \geq 0.1$ and $R^2 \geq$ 95th percentile of the shuffled distribution for that cell (i.e. $p \leq 0.05$) for at least 5 consecutive sliding windows (50 ms total step size). Tuning start times were taken as the midpoint of the first sliding window of the first valid tuning interval. Tuning finish times were taken as the midpoint of the final sliding window of the final valid tuning interval. Tuning duration was calculated as the number of 10 ms time bins that comprised all valid tuning intervals.

To test if there was a significant difference in the proportion of cells with at least one valid tuning interval in stationary and locomotion trials we performed a McNemar test.

To compare the timing of visual speed tuning we only considered cells that were tuned in both stationary and locomotion trials.

We tested whether there was a significant effect of behavioural state on tuning start times, tuning finish times and tuning durations using linear mixed-effects (LME) models with Eq. (4), where the response variable *values* was valid tuning start times, finish times or durations.

**Dynamic range.** The dynamic range of a set of responses was calculated over a sliding window (200 ms window size, 10 ms step size) as the minimum firing rate subtracted from the maximum firing rate across mean responses to all visual speeds.

To test if there was a significant effect of behavioural state on the dynamic range of responses we used an LME model as above (Eq. (4)), where the response variable *values* was the mean value of dynamic range over the stimulus period for each cell.

### Neural correlations
**Partitioning correlations into signal and noise.** To estimate pairwise neural correlations we only considered neurons with mean firing rates ≥1 Hz in both stationary and locomotion trials so that we compared the same pairs of neurons between the two states. This resulted in population sizes of $87 \pm 15$ (mean ± SEM). We calculated pairwise correlations using a sliding window (200 ms step size, 10 ms step size), separately for stationary and locomotion trials. For each neuron pair we first calculated the total Pearson's correlation between spike counts across all trials. We then partitioned these correlations into signal and noise. To estimate signal correlations we shuffled trial spike-counts within stimulus conditions and then performed Pearson's correlations on these shuffled spike-counts for each neuron pair. We repeated this process 10 times and took the mean value as the final estimate of signal correlation. We estimated noise correlations by subtracting the signal correlation value from the total correlation value for each neuron pair.

To assess if there was a significant effect of behavioural state on the average amplitude of signal and noise correlations we used a linear mixed effects (LME) model with the formula:

$$values \sim 1 + state + (1|speed) + (1|subject) \quad (6)$$

where the response variable *values* is the mean absolute value of pairwise signal or noise correlations during the stimulus period for each neuron pair, 1 is a fixed intercept, state is the fixed effect term coding for whether a response was obtained from stationary (0) or locomotion (1) trials, (1|*speed*) and (1|*subject*) terms are random intercepts grouped by stimulus speed and subject.

**Relationship between signal and noise correlations.** We assessed the relationship between signal and noise correlations using linear regression, separately for stationary and locomotion trials. For each session and time window we fit a polynomial of the form $y = mx + c$, where $x$ are signal correlations, $y$ are noise correlations, $m$ is the slope coefficient and $c$ is an offset. We used $m$, the slope coefficient, to quantify the relationship between signal and noise correlations. To assess if there was a significant effect of behavioural state and significant interaction between behavioural state and time on the slope of the signal to noise regression we used a mixed-effects ANOVA with the formula:

$$values \sim time * state * subject \quad (7)$$

where the response variable *values* are regression slope coefficients, *time* and *state* are fixed effects terms and *subject* is a random effects term.

**Stability of the structure of pairwise correlations.** To assess the stability of the structure of pairwise correlations we generated pairwise correlation matrices $C_{N \times N}$ (where $N$ is the number of neurons) for each time window, separately for signal and noise correlations. Each

element of $C$, $C_{ij}$, is the pairwise correlation between neurons $i$ and $j$. We then computed Pearson's correlation between all combinations of correlation matrices obtained from different time windows, producing $T \times T$ temporal correlation matrices, where $T$ is the total number of time windows. Each element of these matrices was Pearson's correlation value between two pairwise correlation matrices.

Because we used a sliding window (200 ms window size; 10 ms step size) to calculate correlations we performed statistical analysis on temporal correlation values for non-overlapping, neighbouring time windows by taking the appropriate diagonal of the temporal correlation matrices. To assess if there was a significant effect of behavioural state and significant interaction between behavioural state and time on the stability of the structure of pairwise correlations we used a mixed-effects ANOVA with Eq. (7) where the response variable *values* were temporal correlation values. We confined our analysis to the stimulus period (midpoints between neighbouring time windows ranged from $t = 0$ ms to $t = 800$ ms).

For visualisation of the pairwise correlation matrices in Fig. 4a, we used hierarchical clustering to order neuron pairs. We first computed a dissimilarity matrix as 1 minus the correlation values. We then sorted the neuron pairs using the Matlab functions *linkage* (using the unweighted average distance) and *dendrogram*. The ordering was generated for the correlation matrix obtained at $t = 800$ ms and applied to the correlation matrices shown.

### Factor analysis
We obtained smooth, single-trial latent trajectories of population activity using Factor Analysis (FA), a linear dimensionality reduction method. We opted to use FA over Gaussian-Process Factor Analysis[105], for example, so that we could retain control over temporal smoothing. We used a linear dimensionality reduction method to aid interpretation of the temporal dynamics of resultant population trajectories. We performed FA on responses to all 12 conditions together (6 speeds × 2 behavioural states), for each subject, so that we could compare population trajectories between behavioural states and across stimuli (e.g. Supplementary Fig. 7).

FA is defined as:

$$x \sim N\left(\mu, LL^{T} + \Psi\right) \quad (8)$$

where $x$ ($n \times 1$) is vector of spike counts from $n$ neurons; $\mu$ ($n \times 1$) is a set of mean spike counts from the same $n$ neurons; $L$ ($m \times n$) is the loading matrix which maps the $m$-dimensional latent variable to the spike counts of $n$ neurons and $\Psi$ ($n \times n$) is a diagonal matrix of independent neuron variance. We estimated $\mu$, $L$ and $\Psi$ using expectation-maximisation[106], with code modified from the DataHigh Matlab toolbox[107]. FA partitions the shared covariance of the spike counts of a population of simultaneously recorded neurons from the independent spike count variance of individual neurons[105,108]. The reduced-dimensional latent factor space obtained using FA therefore represents the shared population variance of a population of simultaneously recorded neurons.

To obtain latent factors of shared population activity, we first binned the spike times of each neuron into 10 ms bins (from 200 ms pre-stimulus onset to 800 ms post-stimulus offset; 2000 ms total). We excluded neurons with a mean smoothed firing rate <1 Hz, resulting in population sizes of $109 \pm 16$ (mean ± SEM). We square root-transformed[105] and smoothed the resultant vectors using a Gaussian kernel with a 35 ms standard deviation, the same kernel we used for PSTH responses. We then $z$-scored these smoothed firing rates to prevent latent factors from being dominated by a small number of high firing rate cells. We obtained similar results irrespective of whether we performed $z$-scoring or not. To determine the dimensionality, $m$, of the latent variable for each subject we used 3-fold cross-validation to find the value of $m$ which maximised the likelihood of the data[108]. We then

obtained a final FA model for each subject by fitting an $m$-dimensional latent factor model.

**Distance measures.** We analysed the paths taken by trial-averaged population trajectories using an $m$-dimensional Euclidean space, where each dimension of the space corresponded to a latent factor.

We determined how direct trial-averaged population trajectories were between steady states by calculating a distance ratio of the cumulative distance travelled by a trajectory between two steady-states, divided by the direct Euclidean distance between the same two points:

$$Cumulative\ distance = \sum_{t=T_{start}+1}^{T_{end}} ||x_t - x_{t-1}||$$
$$Direct\ distance = ||x_{T_{end}} - x_{T_{start}}||$$
$$Distance\ Ratio = \frac{Cumulative\ Distance}{Direct\ Distance} \quad (9)$$

where $x_t$ is position of the $m$-dimensional population trajectory at time $t$, $T_{start}$ is the time at which the population trajectory left the initial steady-state and $T_{end}$ is the time at which the population trajectory reached the final steady-state.

For the stimulus onset period $T_{start}$ was defined as $t = 0\ ms$ (stimulus onset) and for the stimulus offset period $t = 1000\ ms$ (stimulus offset). To determine the time at which population trajectories reached the final steady-state, $T_{end}$, we first defined the size of the steady state as:

$$SS_{size} = \max\left\{ \sum_{t=ss_{start}}^{t=ss_{end}} ||x_t - \bar{x}|| \right\} \quad (10)$$

where $x_t$ is the position of the population trajectory at time $t$, $ss_{start}$ and $ss_{end}$ are the start and end time points of the steady ($ss_{start} = 500\ ms$ and $ss_{end} = 1000\ ms$ for stimulus onset; $ss_{start} = 1500\ ms$ and $ss_{end} = 1800\ ms$ for stimulus offset), $\bar{x}$ is the mean position of the trajectory during the steady state.

We then defined $T_{end}$ in Eq. (9) as the time point at which the population trajectory was closer to the centre of the steady state, $\bar{x}$, than $SS_{size}$ for $\geq 10$ consecutive time windows (200 ms sliding window, 10 ms step size). By definition $T_{end} \leq ss_{start}$.

To test if behavioural state had a significant effect on these 3 distance measures we used linear mixed effects models with Eq. (6), where the response variable *values* were *Cumulative distance*, *Distance travelled* or $log_2$(*Distance Ratio*).

**Speed and acceleration of population trajectories.** To calculate the speed of population trajectories we took the derivative with respect to time of distance travelled. We then divided this value by the number of latent factors. To calculate the acceleration of population trajectories we took the derivative with respect to time of trajectory speed.

To test if behavioural state had a significant effect on max speed during the stimulus onset and offset period we used linear mixed effects models with Eq. (6), where the response variable *values* is the max trajectory speed in either the stimulus onset or stimulus offset period.

We calculated the relative spectral power of trajectory accelerations by generating power spectrum of acceleration vectors in the range 0–20 Hz and dividing these spectrums by their mean value. This produced a bimodal distribution of relative power with peaks greater than and less than 6 Hz. To determine if relative power differed in these two frequency bands between behavioural states we performed a paired $t$-test on the relative power in the low frequency range after testing that the differences in pairs were normally distributed using the Anderson-Darling test.

**Angle of approach of population trajectories.** To calculate the angle of approach of population trajectories we first defined a reference vector. We defined the reference vector as the direct path that joined the mean population trajectory position in an initial and final steady-state:

$$reference\ vector = \bar{x}_{[ss_{final}]} - \bar{x}_{[ss_{initial}]} \quad (11)$$

where $\bar{x}_{[t]}$ is the mean position of the population trajectory in the time interval $t$. For stimulus onset, $ss_{initial}$ was defined as the pre-stimulus period $t = -200\ to\ 0\ ms$ and $ss_{final}$ was defined as the stimulus period $t = 500\ to\ 1000\ ms$. For stimulus offset $ss_{initial}$ was defined as the stimulus period $t = 500\ to\ 1000\ ms$ and $ss_{final}$ as the post-stimulus period $t = 1500\ to\ 1800\ ms$.

We then calculated a vector that defined the instantaneous position of the population trajectory relative to the final steady state:

$$population\ trajectory\ position(t) = \bar{x}_{[SS_{final}]} - x(t) \quad (12)$$

We then calculated the instantaneous angle of approach of the population trajectory relative to the reference vector:

$$\theta(t) = cos^{-1} \frac{a \cdot b(t)}{|a||b(t)|} \quad (13)$$

where $a$ and $b$ are the *reference vector* and *population trajectory position(t)* defined above.

Changes in the angle of approach therefore capture angular deviations away from the direct path between steady-states, as long as they are not orthogonal to it.

We calculated the cumulative angular deviation of a population trajectory by summing the absolute difference in the angle of approach between neighbouring time bins:

$$Cumulative\ angular\ deviation = \sum_{t=0}^{T} \theta_{t+1} - \theta_t \quad (14)$$

To test if the cumulative angular deviation differed significantly between behavioural states we used a linear mixed effects model with the Eq. (6), where the response variable *values* was the cumulative angular deviation of a given population trajectory.

**Population trajectory tangling.** We calculated within-trajectory neural tangling based on previously described methods[49]. We defined neural tangling as:

$$Q_{within}(t) = P_{90}^{(t')}\left\{ \frac{||\dot{x}_t - \dot{x}_{t'}||^2}{||x_t - x_{t'}||^2 + \varepsilon} \right\} \quad (15)$$

where $x_t$ is the position of the population trajectory at time $t$, $\dot{x}_t$ is the velocity of the population trajectory, $|| \ ||$ is the Euclidean norm, $t'$ indexes the time windows over which neural tangling is calculated, and $\varepsilon$ is a small constant which prevents division by 0. For each timepoint we computed the neural tangling between the population trajectory at that time point, $t$, at all other timepoints, $t'$, taking the 90th percentile of these values, $P_{90}^{(t')}\{\}$, as the final value of neural tangling, $Q(t)$. We restricted $t'$ to a 200 ms window around $t$ ($t' \epsilon [t - 100, t + 100]$) to obtain a temporally 'local' neural tangling value. This was to prevent high neural tangling values being obtained from distant time periods, e.g., between stimulus onset and stimulus offset.

To assess if there was a significant effect of behavioural state and significant interaction between behavioural state and time on population trajectory tangling we used a mixed-effects ANOVA with Eq. (7) where the response variable *values* were neural tangling values $Q(t)$.

**Distance between trajectories.** To determine how spread out population trajectory responses to different visual speeds were we calculated the Euclidean distance between trial-meaned trajectories for all combinations of visual speeds and took the mean of these values. We repeated this process for all timepoints to obtain an average inter-trajectory distance over the duration of the response period.

To assess if there was a statistically significant effect of behavioural state on inter-trajectory distance we took the average inter-trajectory distance over the stimulus period and performed a paired $t$-test on these values after testing that the distribution of differences between stationary and locomotion trials was normally distributed using the Anderson-Darling test.

**Shared population activity measures.** We calculated three measures of shared population activity[109]: the percent of total variance that was shared between neurons, the dimensionality of shared variance, and loading similarity of the first latent factor. We calculated these metrics over time using a sliding window (200 ms size, 10 ms step size) by fitting FA models to each time window independently, separately for stationary and locomotion trials. Otherwise, FA models were fit as described above with the exception that we did not smooth spike counts beforehand.

*Percent shared variance* is the average amount of each neuron's variance that can be explained by other simultaneously recorded neurons. It can be calculated by leveraging the fact that FA explicitly partitions neural variance into a shared covariance matrix and an independent diagonal covariance matrix.

$$\% \, shared \, variance \, for \, neuron \, i = \frac{s_i}{s_i + \psi_i} \cdot 100\% \qquad (16)$$

where $s_i$ is the $i$th value of the shared covariance matrix diagonal and $\psi_i$ is the $i$th value of the independent covariance matrix diagonal.

*The dimensionality of shared variance* is the optimal number of dimensions needed to describe shared population activity. We estimated the dimensionality of shared population activity based on previously described methods[108]. We first used cross-validation to find the dimensionality, $m$, that maximised the likelihood of data as described above. We then estimated the final dimensionality as the smallest number of dimensions, $m_{opt}$, required to explain 95% of the shared population variance in the $m$-dimensional model.

*The loading similarity of the first latent factor* describes how similar neural weights are for that factor. It is bound between 0 and 1 such that if all weights are the same, loading similarity = 1, and converges to 0 for weights that are as different as possible. We calculated it as in ref. 109:

$$loading \, similarity(u_k) = 1 - \frac{var(u_k)}{\frac{1}{n}} \qquad (17)$$

where $u_k$ is the unit vector of neuron weights for latent factor $k$, $var(u_k)$ denotes taking the variance and n is the number of neurons contributing to the latent factor (i.e., the number of elements in $u_k$).

To determine the relative contributions of noise and signal correlations to shared population activity measures we computed them separately for (1) intact correlations and (2) disrupted noise correlations by shuffling trials within stimulus conditions.

To assess if behavioural state had a significant effect on any of these metrics (with correlations intact or disrupted) we used a mixed-effects ANOVA with Eq. (7) where *values* were *shared variance, shared dimensionality* or *loading similarity*. To test whether there was a significant interaction between behavioural state and correlation status (intact or disrupted) we performed a separate ANOVA with *correlation status* as an additional fixed effects term.

To estimate % values for the contribution of signal correlations to each population measure we calculated what fraction of values

obtained from intact population activity (signal and noise correlations) was obtained from disrupted population activity (just signal correlations). For loading similarity, this value could occasionally be greater than 1 (<5% of values from stationary trials and <15% of values from locomotion trials) due to the loading similarity being higher for disrupted population activity. We therefore bound all values in the interval [0 1].

**Population trajectory response similarity.** To determine how similar the shapes of population trajectory responses were to different stimulus speeds and in different behavioural states we used Procrustes analysis[110] (Matlab function *procrustes*). Procrustes analysis finds a linear transformation (translation, reflection, orthogonal rotation and isomorphic scaling) that aligns a source shape $Y$ to a target shape $X$ by minimising the squared distance between them. Importantly, the linear transformation preserves the internal configuration of shapes.

The transformed response of $Y$ can be described by:

$$Z = bRY + c \qquad (18)$$

where $Z$ is the transformed response of $Y$ that is optimally aligned with $X$, $b$ is an isomorphic scaling factor, $R$ is an orthonormal matrix specifying a rotation/reflection and $c$ is a translation vector.

After finding the optimal alignment between two population trajectory responses, we calculated the Procrustes distance, $d$, between them. $d$ measures the dissimilarity between the two aligned responses and is bounded in [0,1]. It is calculated as the sum of squared differences between corresponding points in the target response $X$ and aligned response $Z$, normalised by the scale of $X$. The scale of $X$ is calculated as the sum of squared elements of a zero-centred $X$. Because Procrustes distance is a measure of dissimilarity, to obtain a measure of response similarity we simply subtracted $d$ from 1. This final measure of response similarity tended to 1 for maximally similar aligned responses and to 0 for maximally dissimilar aligned responses.

We performed Procrustes analysis on population trajectories during the stimulus onset period ($t = 0$ to 300 ms). We included $m_{opt}$ dimensions of population trajectories after estimating the dimensionality of shared population variance as described above.

## Stimulus decoding

**Independent-neuron decoding of binned spike counts.** We decoded visual speed from binned spike counts using regularised Linear Discriminant Analysis (LDA; Fig. 2h). LDA was performed using the Matlab function *fitdiscr* to fit a *ClassificationDiscriminant* object. Matlab uses LDL decomposition to compute the inverse of a positive semi-definite matrix. To prevent 0 variance predictors, we only included neurons with a mean firing rate ≥1 Hz. This resulted in population sizes of $110 \pm 16$ (mean ± SEM). We performed LDA independently over time using a sliding window (100 ms window size, 10 ms step size) to measure decoding performance over time. We used a diagonal covariance matrix to force the decoder to ignore correlations between neurons.

To train and test decoders we used 3-fold cross-validation whereby decoders were trained on mean binned spike counts from a random 2/3 of trials (split equally between stimulus conditions) and tested on the remaining 1/3. For each cross-validation iteration we repeated this process 3 times with a unique set of test trials each time. We repeated this entire process 5 times for each time window using different random subsets of training and testing trials.

To limit overfitting, within each of these cross-validations we used a further 5-fold cross-validation on training trials to optimise $\delta$, a hyperparameter that acts as a coefficient threshold such that if a neuron (predictor) had an LDA coefficient $<\delta$ it would be ignored as a predictor. We used the 'expected-improvement-per-second' algorithm within Matlab to perform this hyperparameter optimisation.

We evaluated decoder performance as the fraction of held-out test trials correctly predicted.

To assess if there was a significant effect of behavioural state and significant interaction between behavioural state and time on stimulus decoding using binned spike counts we used a mixed-effects ANOVA with Eq. (7) where the response variable *values* was decoding performance values.

**Decoding with intact and trial-shuffled population activity.** To examine how noise correlations influenced stimulus decoding (Supplementary Fig. 8) we performed regularised LDA separately using intact binned spike counts or binned spike counts from trials that had been randomly shuffled independently for each cell within stimulus conditions to disrupt noise correlations.

Decoding was similar to the independent-neuron decoding of binned spike counts described above except the decoder was allowed to estimate the full covariance matrix, allowing it to take into account correlations between neurons. We performed LDA independently over time using a sliding window (100 ms window size, 20 ms step size) and restricted our analysis to the stimulus period.

To train and test decoders we used 3-fold cross-validation whereby decoders were trained on mean binned spike counts from a random 2/3 of trials (split equally between stimulus conditions) and tested on the remaining 1/3. For each cross-validation iteration we repeated this process 3 times with a unique set of test trials each time. We repeated this entire process 3 times for each time window using different random subsets of training and testing trials.

To limit overfitting when training decoders we used 5-fold cross-validation to optimise two hyperparameters: $\delta$, a hyperparameter that acts as a coefficient threshold such that if a neuron (predictor) had an LDA coefficient $<\delta$ it would be ignored as a predictor; and $\gamma$, a hyperparameter that controls the weighting of the full versus diagonal covariance matrix:

$$\Sigma_\gamma = (1 - \gamma)\Sigma + \gamma \, diag(\Sigma) \qquad (19)$$

To calculate the fractional change in performance between trial-shuffled and intact population activity we used the following equation:

$$\Delta Decoding \, Performance = \frac{Performance_{shuffled} - chance}{Performance_{intact} - chance} \qquad (20)$$

where *chance* is the chance performance of a decoder randomly guessing which of the six visual speeds was presented (i.e., 1/6).

**Cross-time decoding of binned spike counts.** To perform cross-time decoding (Fig. 5d, e) we trained and tested LDA decoders in non-overlapping 100 ms windows. To prevent 0 variance predictors, we only included neurons with a mean firing rate ≥1 Hz. For each 100 ms time window we trained a decoder on binned spike counts using half the available trials (split evenly between stimulus conditions, i.e. 2-fold cross-validation). We then used that trained decoder model to predict visual speed from the remaining trials (test trials) in all 100 ms time windows. We repeated this entire process 5 times, taking the mean of these repeats as the decoding performance. As a result, for each 100 ms time window we obtained the performance of a decoder trained in that time window for all time windows.

To limit overfitting when training decoders we used 5-fold cross-validation to optimise two hyperparameters: $\delta$, a hyperparameter that acts as a coefficient threshold such that if a neuron (predictor) had an LDA coefficient $<\delta$ it would be ignored as a predictor; and $\gamma$, a hyperparameter that controls the weighting of the full versus diagonal covariance matrix.

To determine the relative performance of decoders trained in a specific time window we compared its performance to decoders trained and tested in the same time windows (i.e. training window = testing window):

$$Relative \, Performance \, (t) = \frac{\left(\frac{\sum_{i=1}^{T} Perf(train = t, test = i)}{T}\right) - chance}{\left(\frac{\sum_{i=1}^{T} Perf(train = i, test = i)}{T}\right) - chance} \qquad (21)$$

where $t$, is the 100 ms window that a given decoder was trained in, $T$ is the number of non-overlapping 100 ms time windows in the stimulus period and is indexed by $i$, *Perf* is the performance of a decoder with a given train training and testing window and chance is the chance performance of a decoder randomly guessing which of the six visual speeds was presented (i.e., 1/6).

To assess if there was a significant effect of behavioural state on relative decoding performance for decoders trained in the time window ($t = 100$–$200$ ms) we performed a paired t-test after testing the differences in values between stationary and locomotion trials were normally distributed using the Anderson-Darling test.

**Random subsampling of neurons for decoding with multiple pop sizes.** When we decoded using different population sizes we randomly sampled neurons from all neurons with a mean firing rate ≥1 Hz, independently for each repetition. The number of repetitions was equal to the number of available neurons divided by 2. For computational efficiency we then rounded this up to the number of parallel workers available (18). We used this high number of repetitions to ensure that we sufficiently sampled the full distribution of decoding performances for each population size.

**Decoding of single-trial population trajectories.** We decoded visual speed from single-trial population trajectories using a sliding window (50 ms window size, 10 ms step size). Decoding was performed independently in each time window using LDA. In each time window we performed 3-fold cross-validation, where a random 2/3 of trials were assigned as training trials and the remaining 1/3 as testing trials. To train and test decoders we used the mean $m$-dimensional position (where $m$ is the dimensionality of the latent factor model) of the population trajectories in each time window for all trials.

We repeated this entire process 10 times, with different random sets of training and testing trials, and took the mean decoding performance value for each time window as the final decoding performance. We did not optimise hyperparameters since this made little difference to results.

To assess if there was a significant effect of behavioural state and significant interaction between behavioural state and time on stimulus decoding using population trajectories we used a mixed-effects ANOVA with Eq. (7), where the response variable *values* was decoding performance.

**Frequency spectrum of multi-unit activity**
We calculated the frequency spectrum of multi-unit activity (MUA) using binned spike counts.

For each trial, we binned spike times of individual cells into 10 ms non-overlapping bins from 200 ms pre-stimulus onset to 800 ms post-stimulus offset (2000 ms total) and then calculated the mean spike count across all cells for each bin. We then calculated the power spectrum between 1–10 Hz of the resultant binned MUA vector using the Chronux[111] Matlab function *mtspectrumpb*.

We then calculated the mean spectrum across trials for each condition independently (6 stimulus speeds × 2 behavioural states = 12 conditions) and subsequently calculated the normalised power

spectrum for each condition by dividing by the average power for each given condition.

## Probability density estimates

For visualisation purposes where a large number of data points are plotted (e.g., Fig. 1h) we use scatter plots with individual data points coloured by the estimated probability density of datapoints. The probability density estimates are obtained using the Matlab function *ksdensity*. The range of the colorbar is arbitrary for the purpose of visualisation. Where one colorbar used for multiple plots, the scale is consistent across plots to enable comparison.

## Statistics and reproducibility

A full description of statistical analyses is provided in the supplementary materials. No statistical method was used to predetermine sample size. No data were excluded from the analyses. The experiments were not randomised. The Investigators were not blinded to allocation during experiments and outcome assessment. All statistical tests are two-sided.

## Reporting summary

Further information on research design is available in the Nature Portfolio Reporting Summary linked to this article.

## Data availability

Minimally processed data generated in this study have been deposited on Figshare and are available at https://doi.org/10.6084/m9.figshare.26031226.v1. These files enable reproduction of all figures in this publication using the code linked in the 'Code Availability' section. Source data are provided with this paper.

## Code availability

All original code required to reproduce the figures and analysis in this publication are publicly available at https://github.com/eabhorrocks/Horrocksetal24_TemporalDynamics and https://doi.org/10.5281/zenodo.12801956.

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

## Acknowledgements

This work was supported by The Sir Henry Dale Fellowship from the Wellcome Trust and Royal Society (200501); the Human Frontier in Science Program (RGY0076/2018), Biotechnology and Biological Sciences Research Council grant (R004765 and BB/W01579X/1), Medical Research Council grant (MR/W019914/1), UKRI Frontier Research grants (EP/Y024656/1) to A.B.S.; and Biotechnology and Biological Sciences Research Council studentship to E.H. We thank Saskia de Vries and Josh Siegle for discussions, and Laura Busse and Sylvia Shroeder for comments on the manuscript.

## Author contributions

This work was conceptualised by E.H. and A.B.S.; Data collection was by E.H. and F.R.; Methodology, software and formal analysis were by E.H.; Visualisation and writing by E.H. and A.B.S.; and Supervision and funding acquisition by A.B.S.

## Competing interests

The authors declare no competing interests.
