## [Peer Review File · Nature Communications]

REVIEWER COMMENTS

Reviewer #1 (Remarks to the Author):

This manuscript presents original data and analysis showing how profoundly behavioral context (in this case, a mouse being static versus running on a wheel) affects the dynamic neural population coding of sensory stimuli. While the animal is moving, responses to visual motion stimuli are less transient, more reproducible, encoded more efficiently and more stably at the population level.

By performing factor analysis and measuring population trajectories in latent space, the authors further demonstrate the hallmark of a more robust and “stable” dynamics during periods of running: trajectories are untangled, direct and well separated between trial types. In contrast, in static mice, the initial transient is followed by a period of tangled, oscillatory dynamics and a degradation in coding performance. This effect could be in part mediated by Cholinergic Neuromodulation.

This paper presents a very important analysis of the extreme contextual dependency and flexibility of sensory response dynamics, and relate it to its strong influence on neural coding. It was already known that behavioral context has a profound influence on visual coding performance (as reviewed in the introduction). Moreover, a relationship between dynamic trajectory tangling and stability to perturbation had been suggested in the motor cortex (Russo et al, 2018). This paper goes further by applying similar dynamical concepts to sensory responses, introducing a welcome clarification of the relationship between dynamic features and coding performance. It also reports new intriguing effects, such as a post-transient period of oscillatory dynamics in latent population activity (in the static case) associated with a peak in noise correlation and a degradation in coding efficiency. As such, I strongly recommend it for publication in Nature Communication, with minor revisions.

Main comments:

1 – Motor activity has deep effects on the responses in visual cortex of rodents, even in the absence of visual stimuli. The author should establish that the presentation of visual stimuli have no trivial impact on motor behavior, such as affecting running speed, start or end of running periods. Otherwise, changes in neural dynamics could be artefacts of changes in motor behavior.

2 – The visual stimulus somewhat resembles an optic flow created by self-motion in the mice contralateral visual space. In fact, similar stimuli may be used to give mice running on a wheel the percept of moving in a virtual environment. Importantly, the speed of visual motion could be more or less compatible with the animal running speed in any given trial. A mismatch between the visual motion

and the animal motor behavior (e.g. a prediction error) could trigger a stronger visual response, or vice-versa be suppressed as behaviorally irrelevant (or both at different delays). I would thus suggest an additional analysis to check if there is neural correlates of the match/mismatch between visual and running speed, beyond the static/running contextual differences reported here.

3 – The factor analysis of population responses was performed on all the trial combined. However, the visual responses are stronger, more reproducible and more separable by visual speed during running. I am concerned that only due to these properties, FA may have preferentially picked-up latent dimensions spanned by population trajectories during running, while the dimensions spanned during static periods are not as well represented (at least not in the first few “dominant” factors). Thus, trajectories may appear more tangled, and the dynamics more oscillatory, not because the underlying dynamics is different in nature, but because the latent space is less representative.

A way to control for this would be to check that, when a FA is performed separately on running trials and static trials, trajectories remain more tangled/oscillatory in the static context.

Note: FA performed on all trials, as the authors did, is the most appropriate way of comparing trajectories. Separated FA is only a control to check that the measured differences are interpreted correctly.

4- The term “more stable” is almost systematically used in this manuscript to describe the dynamics of responses during running, based on lower entanglement and more direct trajectories. However, “stability” for dynamical systems is normally used to refer to convergence to attractors. This is very different mathematically, and highly confusing since unstable dynamical systems (without attractors) can be untangled, and vice-versa, trajectories of stable dynamical systems can be tangled (if they are close to certain bifurcations, for example).

It is true that the term “stable” was used similarly (and also confusingly) to describe untangled dynamics in the article cited by the authors (Russo et al, 2018). In this previous study, what was specifically meant was robustness (e.g. resistance to perturbations, reproducibility of the output). Since such analysis is absent here, the author should probably remain more concrete and circumspect in their description, and avoid the term “stable dynamics” whenever possible. If “untangled” becomes too repetitive (or too abstract), other terms might be more appropriate (e.g. more robust, smoother, simpler...). Alternatively, if another mathematical notion of dynamic stability (I could be unfamiliar with) is at play, the proper citations (e.g. a textbook) should be provided. Finally, if the term “stable” is kept for the sake of simplicity, there should be an intuitive description of what it means concretely (and what it does not mean) in the context of this study.

Minor comments:

1 – Factor analysis: please provide the spectrum, e.g. the amount of explained variance as a function of how many factors are included, so one may have an idea of the true dimensionality of the population response.

2—Oscillatory dynamics. Is there a signature of oscillatory dynamics (as observed in latent space) in the raw spike trains? e.g. this could be observed as temporary power increase in certain frequency bands of the MUA/summed neural responses.

3 – Abstract is a bit technical for a non-mathematical readership (“tangled oscillatory dynamics”). However I leave that to the appreciation of the authors, since it is extremely difficult to describe dynamical features both intuitively and succinctly.

Reviewer #2 (Remarks to the Author):

This manuscript addresses a fundamental question in sensory systems neuroscience: How does the temporal dynamics of cortical population activity change with behavioural state? And how these changes in temporal dynamics in turn determine the way cortical neurons represent the arriving sensory inputs. This is a fundamental question as it relates to key perceptual and cognitive concepts such as sensory coding efficiency, adaptation, attention, and prediction.

The authors simultaneously record from hundreds of neurons in mouse V1, and quantify neural population responses to moving dots in the visual field during different behavioural states defined by the degree of locomotion.

Using step by step quantitative methods, the authors characterise how behavioural states shape the temporal profile of evoked responses in single V1 neurons, the magnitude and structure of pairwise neural correlations (both signal and noise correlations) and demonstrate how collectively these factors determine the encoding and decoding of sensory inputs in the mouse visual cortex at various time scales. These analyses reveal that in stationary trials (i.e. inactive behavioural states), decoding of visual speed from neural activity is weak and unstable following stimulus onset and improves only slowly over time. Locomotion (i.e. active behavioural state) enhances the decoding of visual speed with faster, and more stable representations following stimulus onset.

Finally, analysis of population trajectories reveal tangled oscillatory dynamics during inactive behavioural states that become dampened and untangled during active states.

This is an elegant work. I did not identify any issues in the data analysis, interpretation and conclusions. The methods of data collection and quantification meet the highest standards in the field. The figures are well organised and informative. I only have one suggestion. The authors could consider discussing parallels between the findings here and those in other cortical areas and sensory modalities. For example, see "State-dependent changes in perception and coding in the mouse somatosensory cortex" by Lee et al, 2020. There are interesting parallels between the findings here and the way behavioural state enhances the sensory evoked responses and changes the population level correlations in the mouse whisker cortex.

Reviewer #3 (Remarks to the Author):

The authors perform a detailed analysis on tuning curves, PSTHs, signal and noise correlations, and population dynamics on large populations of mouse V1 neurons simultaneously recorded as a function of behavioral state. The main conclusion of the paper is that during locomotion tuning is enhanced, signal correlations are enhanced, noise correlations decrease, stimulus encoding enhances, and there is less entangling of neural population trajectories, compared to more stationary conditions of the mice. The results are interesting on the face of the largescale number of single neurons recorded. However, the paper lacks a clear hypothesis beyond that locomotion enhances sensory processing, and it reads as a very descriptive paper. Most of the results are kind of expected based on previous results showing that during locomotion there is enhanced encoding of sensory stimuli (Kafashan et al, Nat Comm, 2021) and several other papers already quoted in the manuscript. The population trajectory analysis is the most novel analytical aspect of the paper, and the result that there is larger separation and less entangling in factor space during locomotion vs stationarity is interesting. Given all the theory it has been developed on information encoding and decoding, the paper lacks a clearly novel contribution, so it might be more suitable for a more specialized journal.

Major Comments:

The stimulus set choice is quite unclear. Why only speed has been chosen to vary, and not

motion direction as well? Why only the nasal-temporal motion direction has been chosen?

More discussion about the stimulus set choice would be relevant.

The definition of locomotion and stationary trials seems to be quite arbitrary. In a 1s trial, in a locomotion trial there could be periods close to stationarity, close 0.5cm/s. How do the main results vary if these definitions are modified?

In the stimulus decoding analysis some details seem to be missing: What is the size of the population being used? In LDA, how the inverse covariance matrix of the neural responses is computed in high dimensions?

There are some relevant papers missing related to the current work:

In (Moreno et al, Nat Neuroscience, 2014) it is shown that very small correlations, called differential correlations, can affect information. It is unclear whether these correlations can still exist under locomotion. As no behavioral thresholds are measured, it is unclear the extent to which the results will extend to behavior in terms of performance.

A possible direction, also explored in Kafashan et al, is to study how decoding performance scales with population size included in the decoder and compare between locomotion and stationary states. Is there any sign of saturation of information?

Minor:

In “Sensory neurons often respond to stimuli with transient onset responses which later settle into a sustained response during sensory stimulation” it is unclear to me what the main references are for this. There is some evidence with some neural responses are very transient (Ray and Maunsell, Plos Biology, 2011).

In “indicates that neural population dynamics are unstable during stationary states” the term “unstable” seems to be used in a way that is different from the standard dynamical systems usage. Do the authors mean “more variable” or “irregular”. Stability analysis would require a different type of analysis that has not been used here.

In panel b Fig. 2, or something like that, color codes for locomotion (red) can be reminded.

Reference

Kafashan, M., Jaffe, A.W., Chettih, S.N. et al. Scaling of sensory information in large neural populations shows signatures of information-limiting correlations. Nat Commun 12, 473 (2021).

<https://doi.org/10.1038/s41467-020-20722-y>

Dear Editor and Reviewers,

We thank the reviewers for their supportive and constructive feedback. We have made several changes in the revised manuscript to address all the points raised and we believe this has further strengthened the manuscript. We summarise our major changes here and more detailed point-by-point responses further below:

- We ran additional analyses to confirm that our results are robust to different criteria for defining stationary and locomotion trials, and cannot be explained by other movement related effects.
- Based on points raised by Reviewer 3, we have tested for saturation of information during stationary and locomotion behavioural states and also confirmed our results using a range of population sizes.
- We re-worded our discussion of population trajectory tangling, avoiding the use of the word 'stability' to prevent confusion with the term as used in dynamical systems theory.
- We expanded our discussion to include how our findings might relate to other brain areas and sensory modalities.

Thank you again for consideration of our revised work and we look forward to your reply.

Reviewer #1 (Remarks to the Author):

This manuscript presents original data and analysis showing how profoundly behavioral context (in this case, a mouse being static versus running on a wheel) affects the dynamic neural population coding of sensory stimuli. While the animal is moving, responses to visual motion stimuli are less transient, more reproducible, encoded more efficiently and more stably at the population level.

By performing factor analysis and measuring population trajectories in latent space, the authors further demonstrate the hallmark of a more robust and “stable” dynamics during periods of running: trajectories are untangled, direct and well separated between trial types. In contrast, in static mice, the initial transient is followed by a period of tangled, oscillatory dynamics and a degradation in coding performance. This effect could be in part mediated by Cholinergic Neuromodulation.

This paper presents a very important analysis of the extreme contextual dependency and flexibility of sensory response dynamics, and relate it to its strong influence on neural coding. It was already known that behavioral context has a profound influence on visual coding performance (as reviewed in the introduction). Moreover, a relationship between dynamic trajectory tangling and stability to perturbation had been suggested in the motor cortex (Russo et al, 2018). This paper goes further by applying similar dynamical concepts to sensory responses, introducing a welcome clarification of the relationship between dynamic features and coding performance. It also reports new intriguing effects, such as a post-transient period of oscillatory dynamics in latent population activity (in the static case) associated with a peak in

noise correlation and a degradation in coding efficiency. As such, I strongly recommend it for publication in Nature Communication, with minor revisions.

We thank the reviewer for their positive comments.

Main comments:

[Reviewer 1, Comment 1 (R1C1)]

1 – Motor activity has deep effects on the responses in visual cortex of rodents, even in the absence of visual stimuli. The author should establish that the presentation of visual stimuli have no trivial impact on motor behavior, such as affecting running speed, start or end of running periods. Otherwise, changes in neural dynamics could be artefacts of changes in motor behavior.

We thank the reviewer for raising an important point about the effects of motor activity on neural activity in visual cortex. Below we present our controls for two types of motor activity, locomotion and eye movements, both of which show that potential changes in motor activity with visual stimuli could not explain our results. We refer to controls for motor activity in **lines 147-150** of the main text.

Locomotion

Only one of five mice (mouse 3) exhibited some modulation of locomotion speed on stimulus presentation, increasing its speed during stimulus presentation (Figure R1a). To control for this, we reanalysed our key results excluding this mouse and found that it did not change our results (**Figure R1**).

Figure R1: Mean locomotion speeds over the response period and the effect of removing subject 3.

a) Trial-averaged wheel speeds for each stimulus condition during locomotion trials. b) Mean reliable PSTHs (related to Figure 1e) with all subjects included (Left panel; 'Original') and with subject 3 excluded (Right panel, 'Subject 3 removed'). c-f) Same as (b) for sustainedness index (c, related to Figure 1h), independent-neuron decoding (d, related to Figure 2h), signal and noise correlations (e, related to Figure 3b), and latent factor stimulus onset distance ratios (f, related to Figure 5d).

We also repeated many of our analyses using two alternative criteria to classify stationary and locomotion trials: using a stricter version of our original criteria or by defining continuous locomotion epochs using a changepoint analysis (based on Lohani et al., 2022; Nat. Neurosci.). Our findings were not affected by using these different behavioural state criteria. This analysis is described in more detail in response to reviewer 3's comments (**R3C2**).

Eye movements

To control for potential effects of eye movements, we identified trials with eye movements and excluded them from our analyses. The effects of behavioural state on PSTH responses persisted - suggesting our results cannot be trivially explained by eye movements. We refer to this in results, **lines 148-150**. The methods for this analysis are described in more detail below.

To detect eye movements we used an outlier detection method. We first calculated the maximum pupil speed in each trial to create a distribution. We then classified a trial as containing eye movements if this maximum speed was ≥ 2.5 median absolute deviations faster than the median. We consider this to be an aggressive threshold that should detect all clear eye movements (based on visual inspection of eye movement traces - see **Figure R2a**). This method removed under 20% of the trials (Stationary: 19+4%, Locomotion: 16+5%, similarly across behavioural states - based on a paired t-test, $p=0.23$, $n=5$ subjects).

Excluding trials with eye movements did not have any clear influence on the PSTH response dynamics between stationary and locomotion states (Figure R2b,c compared to Figure 1e,h).

Figure R2: Removing trials with eye movements did not affect the differences between PSTHs in stationary and locomotion trials.

a) Example random subset of 3 trials (for visualisation purposes) which were classified as containing eye movements (Left panel) and not containing eye movements (right panel). **b)** Mean PSTHs for stationary (black) and locomotion (red) states for trials without eye movements (related to Figure 1e). **c)** scatter density plot of sustainedness index for paired reliable responses for trials without eye movements (related to Figure 1h).

(R1C2)

2 – The visual stimulus somewhat resembles an optic flow created by self-motion in the mice contralateral visual space. In fact, similar stimuli may be used to give mice running on a wheel the percept of moving in a virtual environment. Importantly, the speed of visual motion could be more or less compatible with the animal running speed in any given trial. A mismatch between the visual motion and the animal motor behavior (e.g. a prediction error) could trigger a stronger visual response, or vice-versa be suppressed as behaviorally irrelevant (or both at different

delays). I would thus suggest an additional analysis to check if there is neural correlates of the match/mismatch between visual and running speed, beyond the static/running contextual differences reported here.

This is an interesting point. We performed some additional analyses to detect whether mismatch-type responses are potentially present in our data. We focused on single-neuron PSTH responses since this has been the primary focus of previous work in this area (e.g. Keller et al., Neuron, 2012; Muzzu & Saleem, Cell Reports, 2022). We present this analysis below (**Figure R3**) and as an additional supplementary figure (**Supplementary Figure 5**). We also refer to it in results, **lines: 150-151**.

We first note that prediction error calculation requires the animal to learn an expectation of the sensorimotor environment, usually developed by experiencing (1) closed-loop stimuli coupled animal movement and/or (2) an explicit geometry of the stimulus to create a 'distance heuristic' i.e. the visual speed of a texture expected based on the animals locomotion speed. In our paradigm this is underdetermined - the stimulus could be 'further away' and therefore be expected to have a slower visual speed, or vice versa. These two features allow the generation of a sensorimotor expectation. Our group has shown recently (Muzzu & Saleem; Cell Reports, 2022 & 2023) that responses resembling sensorimotor mismatch can be explained by visual response features, especially in open-loop conditions. As our current stimulus paradigm was not designed for it, we do not have key elements to 'explicitly' test for prediction error-type responses. Specifically, we cannot determine which stimulus visual speeds best 'match' animal locomotion speeds, and in turn cannot infer the subjects' prediction. However, the point raised is an intriguing one, and we therefore sought to determine if we could find any evidence of prediction error-type response modulation in our dataset.

Evidence from existing analysis:

We first investigated the stimulus condition that should maximise mismatch between locomotion and stimulus speed, i.e. when the stimulus is stationary ($0^\circ/s$). In this case any locomotion leads to a mismatch from the expectation that the stimulus should move. Previous research (Keller et al, Neuron, 2012 and follow-up studies) reporting sensorimotor mismatch has usually tested such stationary stimuli. As we show in **Supplementary Figure 1**, whilst temporal dynamics of PSTH responses to the $0^\circ/s$ stimulus are more transient in the stationary state, temporal dynamics are modulated similarly to other visual speeds during locomotion, indicating no clear evidence of prediction error specific modulations to the $0^\circ/s$ stimulus.

Additional analysis

We next tested whether there was evidence of an interaction between locomotion speed and stimulus speed with respect to PSTH responses. We partitioned locomotion trials into 'slow' and 'fast' locomotion trials for each subject. We reasoned that if the animal considered the visual stimulus speeds it experienced to range their own locomotion speeds, then 'slow' locomotion trials with fast visual stimuli or 'fast' locomotion trials with slow visual stimuli would correspond to high mismatch - we used this logic to test for mismatch responses (**Figure R3a**).

To partition trials, we found quartiles of mean stimulus period locomotion speed and classified 'slow' trials as the slowest 25% and 'fast' trials as the fastest 25%, to maximise the difference between the conditions. At the population level, we did not observe any clear differences in the temporal dynamics of PSTH responses between 'slow' and 'fast' locomotion trials at different visual speeds (**Figure R3a-c**), indicating that overall, PSTH response dynamics do not strongly vary with locomotion speed.

Interestingly, we observed a small number of single-neurons with locomotion speed-dependent modulation of PSTH responses. These largely manifested as changes in sustained firing rates, however, some cells did show differences that might reflect more clear changes in temporal dynamics (e.g. example cell 1, 32 °/s). However, most single-neuron sets of responses looked very similar between slow and fast locomotion speeds, reflecting the averages in **Figure R3b,d**. The locomotion-speed dependent responses we did observe were heterogeneous across cells - some cells increased firing rates for fast visual speeds when subjects locomoted faster (i.e. an 'integrative'-type response; **Figure R3e**, example cell 1), whilst other cells increased firing rates for fast visual speeds when subjects locomoted slower (i.e. a potential 'prediction error'-type response as reported by Keller et al, 2012, Neuron; **Figure R3e**, example cell 2).

Of note, in our recordings individual mice tended to locomote at similar speeds across trials. It is possible that with a larger range of locomotion speeds, perhaps reflecting extrema of locomotion gaits, we may see more pronounced effects on PSTH response dynamics. We consider this an interesting line of future research.

Figure R3: PSTH responses for ‘slow’ and ‘fast’ locomotion trials

a) Hypothetical cell responses to the different stimulus visual speeds presented during slow (orange) and fast (green) locomotion. “Non-mismatch”-type cell prediction (left panels) is that responses are strongest when the visual speed best matches that expected from locomotion-induced visual flow. “Mismatch”-type cell prediction is that responses are strongest when the visual speed differs from that expected from locomotion-induced visual flow. **b)** Mean reliable responses for ‘slow’ (orange) and ‘fast’ (green) locomotion trials. **c)** sustainedness index for paired reliable responses from slow and fast locomotion trials. **d)** same as a) for each stimulus visual speed. Note that on average, there was no different in the pattern of responses to each stimulus visual speed between slow and fast locomotion trials, reflecting the majority of single neurons. **e)** Example cell responses that showed locomotion speed-dependent patterns of responses to the stimulus visual speeds. Example cell 1 shows an ‘non-mismatch’-type response in that it increases its firing rate for fast stimulus visual speeds when locomotion speed is also fast. Example cell 2 shows a ‘mismatch-type’ response in that it increases its firing rate for fast visual speeds when locomotion speed is slow. Notably, most cell responses did not exhibit strong locomotion speed-dependence. (b-d are shown in **Supplementary Figure 5**)

(R1C3)

3 – The factor analysis of population responses was performed on all the trial combined. However, the visual responses are stronger, more reproducible and more separable by visual speed during running. I am concerned that only due to these properties, FA may have preferentially picked-up latent dimensions spanned by population trajectories during running,

while the dimensions spanned during static periods are not as well represented (at least not in the first few “dominant” factors). Thus, trajectories may appear more tangled, and the dynamics more oscillatory, not because the underlying dynamics is different in nature, but because the latent space is less representative.

A way to control for this would be to check that, when a FA is performed separately on running trials and static trials, trajectories remain more tangled/oscillatory in the static context.

Note: FA performed on all trials, as the authors did, is the most appropriate way of comparing trajectories. Separated FA is only a control to check that the measured differences are interpreted correctly.

We performed the analysis as suggested by fitting factor analysis models separately to stationary and locomotion trials and comparing the resultant trajectories. As can be seen in the figure below, fitting factor analysis jointly (as in our original analysis) or separately for stationary or locomotion trials had little impact on our results. Indeed, most of the side-by-side comparison plots are hard to differentiate.

A slight difference is in the maximum speed reached by trajectories. In our original analysis, trajectories reached faster maximum speeds following stimulus onset. However, when we fit factor analysis models separately to stationary and locomotion trials the maximum speeds are the same. This is likely a trivial result of independent z-scoring spiking data from stationary and locomotion trials, and therefore we do not believe it warrants further investigation.

We have added a reference to this control analysis in the methods (lines 379-380):

“We obtained similar results if we performed FA on stationary and locomotion trials separately (not shown).”

Figure R4: Oscillatory and tangled population dynamics cannot be explained by latent factors being dominated by stronger visual responses in locomotion trials.

a) Population trajectory speed and acceleration obtained by fitting Factor Analysis (FA) on stationary and locomotion trials together (left panels) and separately (right panels). **b-d)** Same as a) for angles of approach (b), population trajectory tangling (c) and stimulus onset period distance ratios (d).

(R1C4)

4- The term “more stable” is almost systematically used in this manuscript to describe the

dynamics of responses during running, based on lower entanglement and more direct trajectories. However, “stability” for dynamical systems is normally used to refer to convergence to attractors. This is very different mathematically, and highly confusing since unstable dynamical systems (without attractors) can be untangled, and vice-versa, trajectories of stable dynamical systems can be tangled (if they are close to certain bifurcations, for example).

It is true that the term “stable” was used similarly (and also confusingly) to describe untangled dynamics in the article cited by the authors (Russo et al, 2018). In this previous study, what was specifically meant was robustness (e.g. resistance to perturbations, reproducibility of the output). Since such analysis is absent here, the author should probably remain more concrete and circumspect in their description, and avoid the term “stable dynamics” whenever possible. If “untangled” becomes too repetitive (or too abstract), other terms might be more appropriate (e.g. more robust, smoother, simpler...). Alternatively, if another mathematical notion of dynamic stability (I could be unfamiliar with) is at play, the proper citations (e.g. a textbook) should be provided. Finally, if the term “stable” is kept for the sake of simplicity, there should be an intuitive description of what it means concretely (and what it does not mean) in the context of this study.

We thank the reviewer for pointing out that the term ‘stable’ can be confusing when considering population dynamics. We agree and have updated the manuscript in multiple places to address this point. We have avoided using ‘stability’ in reference to population trajectories in order to avoid confusion with the term stability as used in dynamical systems theory. We instead use the term ‘tangling’ directly and have added additional discussion of how to interpret population tangling (Results, **lines 453-465**).

Minor comments:

(R1C5)

1 – Factor analysis: please provide the spectrum, e.g. the amount of explained variance as a function of how many factors are included, so one may have an idea of the true dimensionality of the population response.

We plot the cumulative proportion of shared variance explained as a function of the number of latent factors. We have included this in the revised manuscript (**Supplementary Figure 9**). We note that subjects 3 and 5 have a lower dimensionality, which we have identified as being due to lower trial counts (after downsampling to match between conditions) compared to the other subjects (see Figure 3B of Williamson et al., 2016; PLoS Comput Biol.). We have already shown that removing subject 3 does not alter our findings (see response to point 1, **Figure R1**). Similarly, removing subject 5, or subjects 3 and 5 together, from our analysis did not alter our findings.

Figure R5 (Supplementary Figure 9): Spectra of shared variance explained by latent factors

The cumulative proportion of shared variance explained is plotted as a function of the number of latent factors (in descending order of explained variance, such that factor 1 always explains the most variance) for each subject. Inset: # of trials used (after downsampling to match between conditions) to fit factor analysis models. Factor analysis models for subjects 3 and 5 had lower dimensionality due to reduced trial counts.

(R1C6)

2—Oscillatory dynamics. It there a signature of oscillatory dynamics (as observed in latent space) in the raw spike trains? e.g. this could be observed as temporary power increase in certain frequency bands of the MUA/summed neural responses.

We thank the reviewer for this suggestion. We indeed found a signature of oscillatory dynamics in raw spike trains. We calculated spectrograms of the binned (10ms bin size) population spike count (mean across all neurons) for stationary and locomotion trials, and found an increase in relative spectral power around 3~5Hz during stationary trials. This increase was consistent across stimulus speeds and subjects.

We have added the result of this analysis as a supplementary figure (**Supplementary Figure 11**) and in results **lines: 441-444**. We have also added a discussion of this result in the manuscript, **lines:575-582**.

We have also added the relevant methods, **lines 725-737**.

Figure R6 (Supplementary Figure 11): 3~5Hz relative spectral power of mean population spiking activity is reduced during locomotion trials.

Normalised power spectrums of binned (non-overlapping 10ms bins) mean population activity for stationary (black) and locomotion (red) trials. Shaded regions indicate mean \pm SEM across subjects.

(R1C7)

3 – Abstract is a bit technical for a non-mathematical readership (“tangled oscillatory dynamics”). However I leave that to the appreciation of the authors, since it is extremely difficult to describe dynamical features both intuitively and succinctly.

We have updated the abstract to make it less technical.

Sophie Deneve

Reviewer #2 (Remarks to the Author):

This manuscript addresses a fundamental question in sensory systems neuroscience: How does the temporal dynamics of cortical population activity change with behavioural state? And how these changes in temporal dynamics in turn determine the way cortical neurons represent the arriving sensory inputs. This is a fundamental question as it relates to key perceptual and cognitive concepts such as sensory coding efficiency, adaptation, attention, and prediction.

The authors simultaneously record from hundreds of neurons in mouse V1, and quantify neural population responses to moving dots in the visual field during different behavioural states defined by the degree of locomotion.

Using step by step quantitative methods, the authors characterise how behavioural states shape the temporal profile of evoked responses in single V1 neurons, the magnitude and structure of pairwise neural correlations (both signal and noise correlations) and demonstrate how collectively these factors determine the encoding and decoding of sensory inputs in the mouse visual cortex at various time scales. These analyses reveal that in stationary trials (i.e. inactive behavioural states), decoding of visual speed from neural activity is weak and unstable following stimulus onset and improves only slowly over time. Locomotion (i.e. active behavioural state) enhances the decoding of visual speed with faster, and more stable representations following stimulus onset.

Finally, analysis of population trajectories reveal tangled oscillatory dynamics during inactive behavioural states that become dampened and untangled during active states.

This is an elegant work. I did not identify any issues in the data analysis, interpretation and conclusions. The methods of data collection and quantification meet the highest standards in the field. The figures are well organised and informative. I only have one suggestion. The authors could consider discussing parallels between the findings here and those in other cortical areas and sensory modalities. For example, see "State-dependent changes in perception and coding in the mouse somatosensory cortex" by Lee et al, 2020. There are interesting parallels between the findings here and the way behavioural state enhances the sensory evoked responses and changes the population level correlations in the mouse whisker cortex.

We thank the reviewer for their positive feedback.

As suggested by the reviewer, we expanded on our discussion (**lines: 681-693**, also shown below) regarding how our findings here may relate to other brain areas and sensory modalities, focusing on other visual cortical areas and mouse primary somatosensory (S1) area. There are indeed many interesting similarities between mouse somatosensory cortex and visual cortex. As yet, there is insufficient evidence as to whether these similarities extend to the flexible population response dynamics we found in mouse V1.

“How might response dynamics flexibly change in other brain areas with behavioural states? Our analysis of Allen Institute’s ‘Visual Coding’ dataset⁵⁵ revealed comparable state-dependent changes in the response dynamics of single-neurons in mouse higher visual cortical areas to those we observed in primary visual cortex. This suggests coordinated behavioural-state related changes in population response dynamics across all visual areas. In somatosensory cortex locomotion is associated with depolarised and less variable membrane potentials, increased amplitude of touch stimulus-evoked single-neuron responses and reduces pairwise noise correlations^{19,92,93}, similar to primary visual cortex. Touch stimuli appear to evoke both transient and sustained response dynamics in mouse somatosensory cortex⁹³, and it would be interesting to investigate their state-dependent temporal dynamics at high temporal resolution. Whether flexible state-dependent response dynamics such as those we have observed in mouse visual cortex are also present in brain areas that encode different sensory modalities remains an important open question.”

Reviewer 3

The authors perform a detailed analysis on tuning curves, PSTHs, signal and noise correlations, and population dynamics on large populations of mouse V1 neurons simultaneously recorded as a function of behavioral state. The main conclusion of the paper is that during locomotion tuning is enhanced, signal correlations are enhanced, noise correlations decrease, stimulus encoding enhances, and there is less entangling of neural population trajectories, compared to more stationary conditions of the mice. The results are interesting on the face of the largescale number of single neurons recorded. However, the paper lacks a clear hypothesis beyond that locomotion enhances sensory processing, and it reads as a very descriptive paper.

We thank the reviewer for their comments.

We would like to highlight that the main focus of the manuscript is related to the timecourses of responses. We have further emphasised this in the text to clarify this focus. We also made text changes, as suggested by the reviewer, to more explicitly state our hypotheses (**lines: 20-25 and 38-43**).

Most of the results are kind of expected based on previous results showing that during locomotion there is enhanced encoding of sensory stimuli (Kafashan et al, Nat Comm, 2021) and several other papers already quoted in the manuscript.

Many of the key results of our work relate specifically to the temporal dynamics of the responses to visual stimuli. We agree that enhanced encoding of visual stimuli during locomotion has been shown before, as we clearly state in the manuscript. However, we would like to clarify that the paper quoted and the others referenced within the manuscript almost exclusively focus on neural activity averaged across the stimulus presentation time or otherwise summarised, e.g. by using the peak of the response. We hope that our changes to the introduction outlined above now make it clearer that our hypotheses are largely related to temporal dynamics of neural responses, and that the outcomes of these hypotheses are not readily predicted from previous literature.

The population trajectory analysis is the most novel analytical aspect of the paper, and the result that there is larger separation and less entangling in factor space during locomotion vs stationarity is interesting.

We thank the reviewer for their comments.

Given all the theory it has been developed on information encoding and decoding, the paper lacks a clearly novel contribution, so it might be more suitable for a more specialized journal.

We have updated our text to more thoroughly reflect the theoretical work developed on information encoding and decoding. We have also performed additional analyses related to this in response to point R3C6.

Major Comments:

(R3C1)

The stimulus set choice is quite unclear. Why only speed has been chosen to vary, and not motion direction as well? Why only the nasal-temporal motion direction has been chosen? More discussion about the stimulus set choice would be relevant.

Overall, it was a combination of (a) having the ability to investigate temporal response dynamics without interference from oscillations driven by sensory stimuli (e.g. the phasis responses to drifting gratings), (b) maximising the number of trials for each combination of stimulus and behavioural condition, and (c) the ability to be able to compare to a pre-existing large dataset from the Allen Institute that additionally spans different visual areas. We added a summary of this to the methods section of the paper (**lines 26-28, 30-35**). We also outline a more detailed reasoning for the choice of visual stimuli and parameters below.

Random Dot vs. Drifting grating:

Firstly, random dot fields do not exhibit any overt oscillatory properties unlike drifting gratings, which could easily mask intrinsic oscillatory neural response dynamics. Secondly, random dot

field stimuli evoke strong and widespread responses from neurons in mouse V1 (based on our own previous experiments and previous literature: see e.g. Dyballa et al., 2018) as well as other visual areas (Allen 'Visual Coding' dataset, see fig S3). Here we chose the specific dot size and densities based on previous literature (Marques et al., 2018; Current Biology, Siegle et al., 2021; Nature) and our own data that shows that these parameter values are suitable for driving strong responses in mouse V1 (mentioned in the methods **lines 26-28**). Finally, unlike drifting gratings, random dot fields do not exhibit a strong gaze direction-dependent phase, increasing our ability to draw conclusions about temporal response dynamics across multiple trials, where gaze direction may be varied.

Visual speed:

Firstly, we hypothesised that during active behavioural states with locomotion, the visual system could adapt to be able to rapidly encode the visual speed of sensory inputs as it is a very behaviourally-relevant sensory variable. A further reason we opted to vary visual speed was to enable comparison with the Allen Institute's 'Visual Coding' dataset where the random dot field stimuli varied primarily along the stimulus dimension of visual speed (referred to in methods, **lines 30-35**).

Drift direction:

We used only one direction of motion primarily to maximise the number of trials available in each experimental condition (2 behavioural states x 6 stimulus speeds = 12 conditions). This was especially important as we downsampled the number of trials we used in our analysis to the minimum available across all experimental conditions within each subject, in order to prevent differences in trial counts across conditions confounding our results. Unfortunately, varying motion direction in addition to stimulus speed would result in a substantially larger number of experimental conditions (e.g. 2 behavioural states x 6 stimulus speeds x 6 motion directions = 72 conditions). As a result, using acute electrophysiology it would be impossible to obtain a sufficient number of trials in each experimental condition to perform meaningful analyses. Therefore, we prioritised increased trial repeats over varying more stimulus features.

We chose the naso-temporal direction of motion as it is one of the directions used in the Allen Institute dataset and as we knew it evoked strong and widespread responses in mouse V1 based on previous literature (Marques et al., 2018; Siegle et al, 2021) and our own pilot data.

(R3C2)

The definition of locomotion and stationary trials seems to be quite arbitrary. In a 1s trial, in a locomotion trial there could be periods close to stationarity, close 0.5cm/s. How do the main results vary if these definitions are modified?

We chose these criteria based on previous literature in the field from multiple groups (e.g. Niell and Stryker 2010; Ayaz et al, 2013; Mineault et al., 2016), as well as our own assessment of locomotion speed traces. In **Figure R7a** we plot a 2D histogram of occurrence based on the parameters of our criteria and as can be seen, the distribution is quite bimodal and our criteria capture the modes of these distributions quite effectively.

However, we agree it is important that our results are robust to the choice of criteria and therefore, as a control, we have reanalysed the results using two additional criteria to categorise trials. The first is similar to our original method but with stricter thresholds (**Figure R7a**, 2nd row), and the second uses a changepoints method (Lohani et al., 2022; Nature Neurosci.; **Figure R7a**, 3rd row; described below). Overall, our results hold irrespective of the criteria used to define locomotion and stationary trials. We elaborate this in more detail below and include these results as **Supplementary Figure 4**. We also refer to this analysis in results, **lines 148-149**.

Control analyses - Alternative behavioural state criteria

We present a comparison of our original and the new ‘stricter’ criteria in Figure R7a. We also show an example locomotion epoch classified using the changepoints analysis.

Alternative locomotion criteria

Original: mean wheel speed > 3cm/s and wheel speed >0.5cm/s for $\geq 75\%$ of 2s trial period.

Stricter: mean wheel speed >3cm/s and wheel speed >0.5cm/s for $\geq 90\%$ of 2s trial period.

Changepoints: a change-point algorithm to detect periods of locomotion based on changes in the standard deviation of the raw rotary encoder data (following methods from Lohani et al., 2022; Nat. Neurosci.). See methods **lines 132-150** for details.

Alternative stationary criteria:

Original: mean wheel speed < 0.5cm/s and wheel speed <3cm/s for $\geq 75\%$ of 2s trial period.

Stricter: wheel speed <0.5cm/s throughout 2s trial period.

Changepoints: same as used in ‘stricter’.

Note that in **Figure R7** where we compare original and stricter criteria, we have plotted a simplified definition of stationary trials using the original criteria for ease of visualisation (we plot proportion of time <0.5cm/s instead of the <3cm/s threshold we used). These two criteria were nearly equivalent in our dataset: trial count for plotted criteria = 97 \pm 1 % (mean \pm SEM) of actual criteria.

Results with different state criteria

None of our main results were affected by changing the criteria we used for defining stationary and locomotion trials. We present a subset of representative results in **Figure R7**.

Our rationale for allowing a small number of time bins with wheel speed <0.5cm/s in our definition of locomotion trials is that many mouse locomotion gaits on a styrofoam wheel produce rhythmic momentary periods of slow wheel speeds, even during extended periods locomotion (**Figure R7a**, 3rd row; see also e.g. Lohani et al., 2022; Nat. Neurosci. Fig 3a). As a result, our original criteria provide a good balance between allowing trials occurring during such locomotion epochs to be correctly classified as locomotion trials and excluding trials in which subjects are actually stationary.

Figure R7: Robustness of results to different behavioural state criteria.

a) Overview of the three different behavioural state criteria that we compared our results across. Top row: 2D histograms of time spent $> 0.5\text{cm/s}$ during the 2s trial periods and the mean locomotion speed during trials, for each subject. Red and blue and blue boxes indicate trials classified as locomotion and stationary respectively. Middle row: same as the top row, except that red and blue boxes indicate the 'strict' criteria. Bottom row: illustration of a locomotion epoch identified using a changepoints analysis (Lohani et al., 2022). Stationary trials were classified using the same criteria as 'strict' for this classification. **b)** Comparison of mean reliable PSTH responses across the three behavioural state criteria. **c-f)** Same as **b)** for distributions of difference in sustainedness index between locomotion and stationary states, where the 3 horizontal red lines indicate the quartiles of each distribution (**c**); mean absolute signal (top row) and noise (bottom row) correlations (**d**); distance ratios for the stimulus onset period of responses (**e**); decoding performance using latent factors (**f**).

(R3C3)

In the stimulus decoding analysis some details seem to be missing: What is the size of the population being used? In LDA, how the inverse covariance matrix of the neural responses is computed in high dimensions?

We thank the reviewer for their careful observations.

We have added the population sizes and further information about how LDA was performed in the Methods section, **lines 607-610** (also see below). We also reviewed the rest of the methods section of the paper carefully, and added additional information where appropriate. These are highlighted as changes in the revision.

To prevent 0 variance predictors, we only included neurons with a mean firing rate ≥ 1 Hz. This resulted in population sizes of 110 ± 16 (mean \pm SEM). We used the matlab function *fitdiscr* to perform regularised LDA. Matlab uses LDL decomposition to compute the inverse of Hermitian matrices.

Additionally, to test that our stability of decoding readout results (Figure 4d-e) were not unduly influenced by issues related to estimating the high-dimensional covariance matrix, we repeated our analysis with different population sizes (n=10, 20, 40, 80). This resulted in similar results regardless of population size (**Figure R8** - additionally see the response to the point **R3C6** below). This analysis is included in **Supplementary Figure 8c**.

(R3C4)

There are some relevant papers missing related to the current work:

In (Moreno et al, Nat Neuroscience, 2014) it is shown that very small correlations, called differential correlations, can affect information. It is unclear whether these correlations can still exist under locomotion.

We have added this reference, as well as Kafahsan et al., to our results, **lines 248** and discussion, **line 625**. We have also made text changes to more thoroughly reflect theoretical work on the influence of neural correlations on population encoding. We also perform some additional related analyses which we outline below in response to point **R3C6**.

(R3C5)

As no behavioral thresholds are measured, it is unclear the extend to which the results will extend to behavior in terms of performance.

We have predicted the impacts of the changes in temporal response dynamics between behavioural states on perception and behaviour in the discussion (**lines 538-542**).

(R3C6)

A possible direction, also explored in Kafahsan et al, is to study how decoding performance scales with population size included in the decoder and compare between locomotion and stationary states. Is there any sign of saturation of information?

We thank the reviewer for this interesting suggestion.

We have performed additional analyses related to this point, where we compare decoding performance for different population sizes using intact population activity or population activity with disrupted noise correlations (by shuffling trials within conditions), for different population sizes.

Our main findings are:

- 1) disrupting noise correlations by shuffling trials results in improvements in linear decoding performance which scale with population size, for both stationary and locomotion states. Although we are aware that this is an imperfect measure, we believe it suggests saturation of information is present in both behavioural states.
- 2) whilst changes in performance were largely constant across time during locomotion trials, they clearly varied over time during stationary trials. Specifically, disrupting noise correlations during stationary trials resulted in improved decoding performance principally between 200~350ms post-stimulus onset.

We have added this result as **Supplementary Figure 8** and refer to it in Results, **lines 268-273**. We have also added discussion of these results with reference to Moreno et al., Kafahsan et al., and others in discussion, **lines: 618-627**. We have also included the Methods, **lines: 634-665**.

We consider this an important direction of future research using experimental designs specifically optimised to test it.

Figure R8: Decoding analysis using different population sizes (Supplementary Figure 8)

a) Mean decoding performance using regularised LDA for 4 different population sizes (10, 20, 40 or 80 neurons), for stationary (black) and locomotion trials (red), with either intact population activity (solid lines) or trial-shuffled population activity (dashed lines) which disrupts noise correlations. Shaded regions indicate mean \pm SEM across individual populations (# pops). **b)** Median Δ decoding performance between intact and shuffled population activity calculated as $(\text{shuffled performance} - \text{chance}) / (\text{intact performance} - \text{chance})$. Errorbars indicate ± 1 z-score of the distribution. Note the consistent increase in performance for trial-shuffled population activity for stationary trials between 0.2~0.4s following stimulus onset. **c)** Related to Figure 4e. Relative decoding performance for decoders trained in the time window 100-200ms following stimulus onset, for 4 different population sizes. Horizontal white lines indicate the median and inter-quartile range of each distribution. Note that relative decoding performance was consistent across population sizes, indicating that cross-time generalisation is higher during locomotion for this training window.

Minor:

(R3C7)

In “Sensory neurons often respond to stimuli with transient onset responses which later settle into a sustained response during sensory stimulation” it is unclear to me what the main references are for this. There is some evidence with some neural responses are very transient (Ray and Maunsell, Plos Biology, 2011).

We agree that single-neuron response dynamics are diverse. We also see very transient responses in our data (**Figure 1d** - reliable stationary responses). We have updated the sentence to better reflect the diversity of single-neuron response dynamics described in the literature and added additional references, **lines: 15-16**.

“Sensory neurons can respond to external stimuli with varied temporal dynamics, often exhibiting transient onset responses which later settle into a sustained response”

(R3C8)

In “indicates that neural population dynamics are unstable during stationary states” the term “unstable” seems to be used in a way that is different from the standard dynamical systems usage. Do the authors mean “more variable” or “irregular”. Stability analysis would require a different type of analysis that has not been used here.

We agree and have updated the manuscript in multiple places to address this point. We have avoided using ‘stability’ in reference to population trajectories in order to avoid confusion with the term stability as used in dynamical systems theory. We instead use the term ‘tangling’ directly and have added additional discussion of how to interpret population tangling - see Results, **lines 453-465**.

(R3C9)

In panel b Fig. 2, or something like that, color codes for locomotion (red) can be reminded.

We have added an additional legend to **Figure 2**.

Reply to Reviewers References

Ayaz A, Saleem AB, Schölvinc ML, Carandini M. Locomotion controls spatial integration in mouse visual cortex. *Curr Biol*. 2013;23(10):890-894. doi:10.1016/j.cub.2013.04.012

Keller GB, Bonhoeffer T, Hübener M. Sensorimotor mismatch signals in primary visual cortex of the behaving mouse. *Neuron*. 2012;74(5):809-815. doi:10.1016/j.neuron.2012.03.040

Lohani S, Moberly AH, Benisty H, et al. Spatiotemporally heterogeneous coordination of cholinergic and neocortical activity. *Nat Neurosci*. 2022;25(12):1706-1713. doi:10.1038/s41593-022-01202-6

Marques T, Summers MT, Fioreze G, et al. A Role for Mouse Primary Visual Cortex in Motion Perception. *Curr Biol*. 2018;28(11):1703-1713.e6. doi:10.1016/j.cub.2018.04.012

Mineault PJ, Tring E, Trachtenberg JT, Ringach DL. Enhanced Spatial Resolution During Locomotion and Heightened Attention in Mouse Primary Visual Cortex. *J Neurosci*. 2016;36(24):6382-6392. doi:10.1523/JNEUROSCI.0430-16.2016

Muzzu T, Saleem AB. Feature selectivity can explain mismatch signals in mouse visual cortex [published correction appears in *Cell Rep*. 2022 Feb 15;38(7):110413]. *Cell Rep*. 2021;37(1):109772. doi:10.1016/j.celrep.2021.109772

Muzzu T, Saleem AB. Redefining sensorimotor mismatch selectivity in the visual cortex. *Cell Rep*. 2023;42(3):112098. doi:10.1016/j.celrep.2023.112098

Niell CM, Stryker MP. Modulation of visual responses by behavioral state in mouse visual cortex. *Neuron*. 2010;65(4):472-479. doi:10.1016/j.neuron.2010.01.033

Siegle JH, Jia X, Durand S, et al. Survey of spiking in the mouse visual system reveals functional hierarchy. *Nature*. 2021;592(7852):86-92. doi:10.1038/s41586-020-03171-x

Williamson RC, Cowley BR, Litwin-Kumar A, et al. Scaling Properties of Dimensionality Reduction for Neural Populations and Network Models. *PLoS Comput Biol*. 2016;12(12):e1005141. Published 2016 Dec 7. doi:10.1371/journal.pcbi.1005141

REVIEWERS' COMMENTS

Reviewer #2 (Remarks to the Author):

The authors have appropriately addressed the point that I raised in my original review. I find the paper well written and highly informative.

Reviewer #3 (Remarks to the Author):

The authors have appropriately addressed all my comments